# UNIP: Rethinking Pre-trained Attention Patterns for Infrared Semantic Segmentation

**Tao Zhang[1,2]** **Jinyong Wen[1,2]** **Zhen Chen[3]** **Kun Ding[1]*** **Shiming Xiang[1,2]** **Chunhong Pan[1]**

[1]MAIS, Institute of Automation, Chinese Academy of Sciences, China
[2]School of Artificial Intelligence, University of Chinese Academy of Sciences, China
[3]School of Automation Science and Electrical Engineering, Beihang University, China
`{zhangtao2021,kun.ding}@ia.ac.cn`

## Abstract

Pre-training techniques significantly enhance the performance of semantic segmentation tasks with limited training data. However, the efficacy under a large domain gap between pre-training (*e.g.* RGB) and fine-tuning (*e.g.* infrared) remains underexplored. In this study, we first benchmark the infrared semantic segmentation performance of various pre-training methods and reveal several phenomena distinct from the RGB domain. Next, our layerwise analysis of pre-trained attention maps uncovers that: (1) There are three typical attention patterns (local, hybrid, and global); (2) Pre-training tasks notably influence the pattern distribution across layers; (3) The hybrid pattern is crucial for semantic segmentation as it attends to both nearby and foreground elements; (4) The texture bias impedes model generalization in infrared tasks. Building on these insights, we propose **UNIP**, a **UN**ified **I**nfrared **P**re-training framework, to enhance the pre-trained model performance. This framework uses the hybrid-attention distillation NMI-HAD as the pre-training target, a large-scale mixed dataset InfMix for pre-training, and a last-layer feature pyramid network LL-FPN for fine-tuning. Experimental results show that UNIP outperforms various pre-training methods by up to **13.5%** in average mIoU on three infrared segmentation tasks, evaluated using fine-tuning and linear probing metrics. UNIP-S[1] achieves performance on par with MAE-L while requiring only **1/10** of the computational cost. Furthermore, UNIP significantly surpasses state-of-the-art (SOTA) infrared or RGB segmentation methods and demonstrates broad potential for application in other modalities, such as RGB and depth. Our code is available at https://github.com/casiatao/UNIP.

## 1 Introduction

Pre-training is essential in computer vision, equipping models with fundamental feature extraction capabilities. Supervised methods (Touvron et al., 2021; 2022) and self-supervised methods, such as contrastive learning (CL) (Chen et al., 2021b; Caron et al., 2021) and masked image modeling (MIM) (He et al., 2022; Fu et al., 2024), have demonstrated great potential in various visual tasks, particularly for small-scale datasets. Infrared images, widely used in road surveillance (Bondi et al., 2020), autonomous driving (Xiong et al., 2021), and unmanned aerial vehicle (Sun et al., 2022), often lack labeled data for tasks like object detection and semantic segmentation (Li et al., 2021a). Therefore, having a strong pre-trained backbone is vital for these data-limited scenarios.

**However, the transfer performance on infrared segmentation of different pre-training methods remains considerably underexplored**. Previous works (Xiong et al., 2021; Chen & Bai, 2023) aim to improve performance by designing specific architectures for infrared segmentation tasks, without assessing the impact of various pre-training methods on model performance. Additionally, mainstream pre-training methods (He et al., 2022; Zhou et al., 2022) usually evaluate performance on large-scale RGB datasets like ImageNet (Deng et al., 2009) and ADE20K (Zhou et al., 2017).

---

*Kun Ding is the corresponding author.

[1]We use the term *method-size* to denote the vision transformer (ViT) of a specific *size* pre-trained by a specific *method*. T, S, B, and L refer to the ViT-Tiny, ViT-Small, ViT-Base, and ViT-Large, respectively.

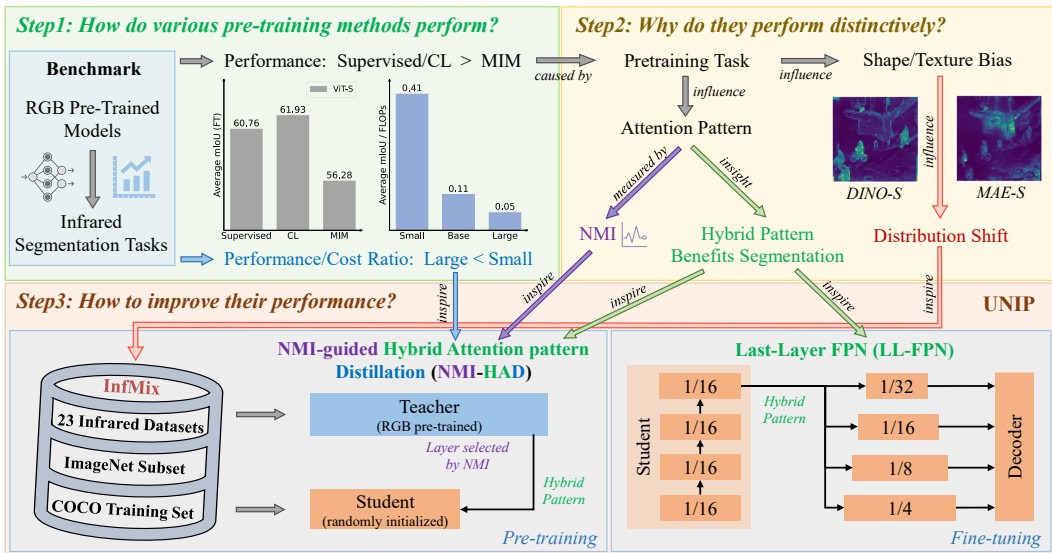

Figure 1: The Chain-of-Thought (CoT) of our work. *Step1* (Sec. 2): We benchmark the infrared segmentation performance of various pre-trained models and derive several insights. *Step2* (Sec. 3): We explore the reasons for the varying behaviors of these models by analyzing the pre-trained attention maps. *Step3* (Sec. 4): Based on these findings, we propose UNIP, a unified framework aimed to enhance the performance of small pre-trained models, focusing on three aspects: the pre-training dataset (InfMix), the pre-training task (NMI-HAD), and the fine-tuning architecture (LL-FPN).

Given the significant domain differences between RGB and infrared datasets, further study is necessary to evaluate the transfer performance of different pre-training methods on infrared visual tasks and the validity of phenomena observed in RGB datasets for the infrared domain.

To this end, we benchmark six popular supervised and self-supervised (CL and MIM) pre-training methods on three infrared semantic segmentation datasets, across different model sizes and evaluation metrics (see Sec. 2, Step1 in Fig. 1). Some valuable phenomena are discovered: (1) The ImageNet accuracy of models does not necessarily correlate with their performance on infrared segmentation tasks; (2) Supervised and CL methods exhibit better generalization than MIM methods, especially for small models like ViT-T and ViT-S; (3) The performance improvement of larger models is marginal compared to the substantial increase in computational cost, making them unsuitable for infrared-related tasks that require fast processing speeds with limited computing resources.

To understand the distinct performance of these methods, we conduct a thorough analysis of attention maps (see Sec. 3, Step2 in Fig. 1). Three attention patterns–*local*, *hybrid*, and *global*–are identified in different layers of pre-trained models. As shown in Fig. 3, *local* patterns focus on nearby tokens[2], while *global* patterns prefer foreground tokens. *Hybrid* patterns attend to both types. The pre-training tasks significantly influence the pattern distributions: **Supervised and CL models exhibit *all* patterns, whereas MIM models show only *local* and *hybrid* patterns. Importantly, the *hybrid* attention pattern is found to be crucial for semantic segmentation as it can effectively capture both local and global information**. To quantitatively distinguish these patterns, we introduce the normalized mutual information (NMI) between query and key tokens as an indicator, which aligns well with pattern distributions. Additionally, we find that the bias towards texture observed in attention maps can exacerbate distribution shifts and hinder model generalization in infrared tasks.

Based on the above analysis, a UNified Infrared Pre-training framework called **UNIP** is proposed to enhance the infrared segmentation performance of small models (see Sec. 4, Step3 in Fig. 1). First, we introduce the **NMI**-guided **H**ybrid **A**ttention pattern **D**istillation (**NMI-HAD**) as the pre-training target, which uses NMI to select the distillation layer and compresses *hybrid* patterns from teacher models to randomly initialized student models. Second, to bridge the gap between pre-training and infrared data and mitigate distribution shifts, we construct a large mixed dataset called **InfMix** as the pre-training dataset. It comprises **859,375** images from **25** datasets, ensuring no overlap with the segmentation datasets used in our benchmark. Third, to utilize *hybrid* patterns in the last layer of

---

[2]In this work, *token* is used to denote the 16×16 patch in the image.

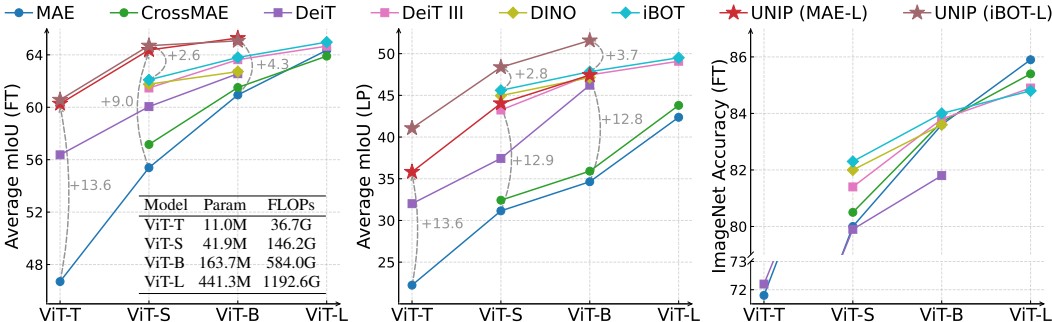

Figure 2: The performance of pre-trained models across various methods and sizes. *Left*: The average fine-tuning (FT) performance on three infrared semantic segmentation datasets, along with the associated computational cost. *Middle*: The average linear probing (LP) performance on three infrared datasets. *Right*: The fine-tuning performance on ImageNet (Deng et al., 2009). The gray dotted lines and corresponding values highlight the performance gains of UNIP over other methods. Detailed results for each dataset are presented in Tab. 11.

distilled modes, we propose the **L**ast-**L**ayer **F**eature **P**yramid **N**etwork (**LL-FPN**) for fine-tuning to enhance performance further. With these enhancements, the average segmentation mIoU of UNIP significantly surpasses their counterparts, as shown in Fig. 2 and Tab. 4. When using MAE-L (He et al., 2022) as the teacher, UNIP achieves improvements of **13.57%** (T), **8.98%** (S), and **4.34%** (B) in fine-tuning, and at least **12.79%** in linear probing. With iBOT-L (Zhou et al., 2022) as the teacher, UNIP-S exceeds iBOT-S by **2.61%** in fine-tuning, while UNIP-B surpasses iBOT-B by **3.74%** in linear probing. Notably, the distilled models even outperform their teacher models across different pre-training methods. UNIP also substantially outperforms other SOTA infrared or RGB segmentation methods, such as TINN (Chen & Bai, 2023) and Mask2Former (Cheng et al., 2022), and exhibits effectiveness and application potential in other modalities like RGB and depth images.

Our main contributions consist of (1) A comprehensive benchmark of six pre-training methods on three infrared semantic segmentation datasets, highlighting several key phenomena; (2) A detailed investigation of pre-trained attention patterns, emphasizing the critical importance of the *hybrid* pattern for semantic segmentation; (3) A unified infrared pre-training framework UNIP, including the NMI-HAD method, the InfMix dataset, and the LL-FPN architecture; (4) Extensive experimental results, demonstrating the effectiveness and efficiency of our method and dataset.

## 2 HOW DO PRE-TRAINING METHODS PERFORM ON INFRARED TASKS?

In this section, we benchmark six pre-training methods on three infrared semantic segmentation datasets and discuss several key phenomena.

### 2.1 INFRARED SEGMENTATION BENCHMARK OF RGB PRE-TRAINED MODELS

**Pre-trained Backbone.** Pre-training of the Vision Transformer (ViT) (Dosovitskiy et al., 2021) has gained widespread attention and demonstrated powerful performance in various fields. Many recent pre-training methods (Touvron et al., 2021; He et al., 2022; Zhou et al., 2022; Oquab et al., 2024) use ViT for experiments, making pre-trained ViT models readily available. Therefore, ViT models of various sizes are set as the evaluation backbone.

**Pre-training Methods.** Both supervised and self-supervised methods are investigated. For supervised approaches, we use **DeiT** (Touvron et al., 2021) and **DeiT III** (Touvron et al., 2022), which perform image classification on ImageNet for pre-training. In self-supervised methods, we study contrastive learning (CL) and masked image modeling (MIM). CL methods like **DINO** (Caron et al., 2021) encourage features from different views of the same image to be close, while keeping features from different images distinct. MIM methods like **MAE** (He et al., 2022) and **CrossMAE** (Fu et al., 2024) focus on reconstructing masked image patches by learning context relations. Although **iBOT** (Zhou et al., 2022) combines CL with masked feature prediction, we classify it as a CL method due to its similar characteristics to DINO. **The above methods are selected because they all pre-train vanilla ViT models on ImageNet**, without additional pre-trained tokenizers like BeiT (Bao et al.,

2022) or MILAN (Hou et al., 2022), or larger datasets like EVA (Fang et al., 2023) and DINOv2 (Oquab et al., 2024). This allows us to focus on the impact of the pre-training tasks alone.

**Evaluation Datasets.** The evaluation is conducted on three infrared semantic segmentation datasets: SODA (Li et al., 2021a), MFNet-T (Ha et al., 2017), and SCUT-Seg (Xiong et al., 2021). Notably, MFNet is an RGB-Thermal paired dataset. The thermal part MFNet-T is used for benchmarks while the RGB part MFNet-RGB is employed in further investigations. Additionally, RGB datasets like ImageNet-1K (Deng et al., 2009) and ADE20K (Zhou et al., 2017) are also used for comparison. Details about these datasets can be found in Appendix C.3.

**Evaluation Metrics.** We employ two metrics: *fine-tuning* (FT) and *linear probing* (LP). FT (Fig. 8a) is the primary metric, where both the pre-trained model and the decoder are tuned with the labeled datasets. In LP (Fig. 8b), only a linear head is updated while all other parameters remain frozen. **Average (Avg) FT or LP performance in subsequent sections denotes the mean mIoU across three infrared semantic segmentation datasets.** More details are available in Appendix C.2.

**Benchmark Results.** In the benchmark, all models are trained for 100 epochs for both evaluation metrics. Typical results are illustrated in Fig. 2, with ImageNet (Deng et al., 2009) fine-tuning performance included for comparison. The complete results of each dataset are detailed in Tab. 11.

## 2.2 WHAT INSIGHTS CAN WE GAIN FROM THIS BENCHMARK?

**The infrared FT performance is strongly positively correlated with LP, but has no clear relationship with ImageNet FT.** Tab. 1 presents the Pearson (Pearson, 1896) correlation coefficients between different metrics. For each metric pair, the coefficients are calculated across six pre-training methods in Fig. 2. Notably, the coefficients between average infrared

Table 1: Pearson coefficients between average FT and other metrics.

| Metric | Small | Base | Large | Mean |
|---|---|---|---|---|
| Avg FT & Avg LP | **0.89** | **0.93** | **0.81** | **0.88** |
| Avg FT & IN1K FT | 0.78 | 0.12 | -0.66 | 0.08 |

LP and FT are close to 1, indicating that models with better LP performance generally exhibit better FT performance. Conversely, the ImageNet FT performance does not consistently correlate with infrared FT results across various model sizes, likely due to domain and task differences. Therefore, using ImageNet accuracy to predict transfer performance on infrared segmentation datasets is not reliable, underscoring the importance of benchmarking on infrared segmentation datasets.

**Supervised and CL methods outperform MIM methods, especially for small models.** As depicted in Fig. 2 and Tab. 11, the performance of supervised and CL methods like DeiT, DeiT III, DINO, and iBOT is similar across both metrics, except for the LP of DeiT-S. For the LP metric, MIM methods of various sizes consistently lag behind supervised and CL methods by a significant margin, matching observations in the RGB domain (He et al., 2022) that MIM representations are less linearly separable. In terms of FT, smaller MIM models (ViT-T, S, and B) still underperform supervised and CL methods, while larger models (ViT-L) are more competitive. For instance, MAE-S is far behind iBOT-S (55.39% vs 62.09%), but MAE-L performs comparably to iBOT-L (64.35% vs 64.97%). As we will discuss in Sec. 3, the discrepancy in the attention pattern distribution and texture bias between different models accounts for their distinct infrared segmentation performance.

**Larger models perform better, but their computational cost increases sharply.** As illustrated in Fig. 2, larger MIM models bring considerable performance gains over smaller models. However, for supervised and CL methods, small models are already well-trained, and the performance improvement from larger models is marginal compared to the significant increase in computational cost. For example, iBOT-L surpasses iBOT-S by only 2.85% (64.97% vs 62.09%), while the parameter count and FLOPs increase by $10.5\times$ (441M vs 42M) and $8.2\times$ (1193G vs 146G), respectively. Given that infrared images are often processed on edge devices with limited computing budgets, using large models to pursue better performance is not cost-effective. Therefore, we believe improving small models is a more effective approach. In Sec. 4, we propose several strategies to elevate the performance of small models to be on par with larger models.

## 3 WHAT MATTERS FOR INFRARED SEMANTIC SEGMENTATION?

To determine which characteristics of the pre-trained models are critical for infrared semantic segmentation, we analyze different models from multiple perspectives.

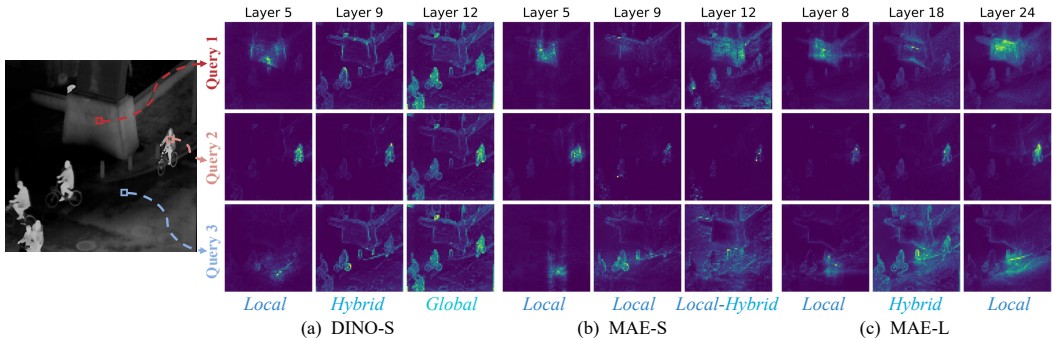

Figure 3: Attention maps for different query tokens in three representative layers. Each query token's attention map corresponds to a row in the attention matrix, averaged over different heads.

### 3.1 THE PRE-TRAINING TASKS INFLUENCE ATTENTION PATTERNS

The self-attention mechanism is a key component of ViT. In semantic segmentation tasks, the spatial interactions between tokens are crucial. Thus, we visualize the attention maps of pre-trained models.

**Attention maps of supervised/CL and MIM methods differ significantly.** As shown in Fig. 3a, DINO-S exhibits three distinct attention patterns: (1) *Local*: In shallow layers (Layer 5), different query tokens focus only on their spatially nearby key tokens; (2) *Hybrid*: In middle layers (Layer 9), query tokens attend to both nearby tokens and foreground tokens; (3) *Global*: In deep layers (Layer 12), different query tokens all focus on foreground tokens with nearly identical attention maps, a phenomenon known as *attention collapse* (Park et al., 2023). This attention pattern distribution is consistent across different sizes in CL and supervised methods, as shown in Fig. 10. However, in MIM methods like MAE, the distribution varies. In MAE-S (Fig. 3b), attention maps are mainly *local*, with slight *hybrid* patterns emerging in deep layers. Conversely, in MAE-L (Fig. 3c), shallow and deep layers exhibit *local* patterns, while middle layers show *hybrid* patterns. CKA (Kornblith et al., 2019) analysis in Appendix D.2 reveals similar phenomena regarding feature representation.

**Differences in attention patterns stem from the pre-training tasks.** CL methods, similar to supervised approaches, treat views from the same image as belonging to the same class. This setup encourages models to focus on foreground tokens, as images in the same class often share similar foreground objects but may differ in background. Consequently, attention maps in later layers present *global* patterns. The *local* and *hybrid* patterns can be regarded as the intermediate states in forming the *global* pattern. This high-level pre-training task causes models of different sizes and methods to have similar pattern distributions across layers. In contrast, pre-training tasks of MIM methods focus on reconstructing features or raw pixels of masked tokens, which is a relatively low-level task relying heavily on spatially nearby tokens. Consequently, models are not compelled to capture global image information, leading small models to primarily exhibit *local* patterns. In larger models like MAE-L, the increased representation capacity allows *hybrid* patterns to spontaneously emerge in the middle layers to capture broader context. In deep layers near the decoder, *local* patterns reappear to support the pre-training task of reconstructing nearby masked tokens.

As a supplement, iBOT exhibits similar patterns with DINO in shallow and middle layers but shows less *attention collapse* in deep layers (see Fig. 10). This can be attributed to that iBOT combines DINO with masked feature prediction (Zhou et al., 2022), which encourages the later layers to leverage spatial information to predict features of masked tokens.

### 3.2 HOW TO QUANTITIVELY IDENTIFY DIFFERENT ATTENTION PATTERNS?

Three distinct attention patterns are qualitatively summarized in Sec. 3.1, prompting the question of whether a metric can quantitatively measure them. Attention distance measures the average distance between the query and key tokens, while attention entropy implies the concentration of the attention distribution. However, both metrics depict the relationship between one query and multiple key tokens and are unable to reflect differences in the attention maps of various queries. **We find that the normalized mutual information (NMI) between query and key tokens is an effective indicator.** The calculation process is elaborated in Appendix D.1. Let $A \in \mathbb{R}^{N \times N}$ denote the attention matrix, where $N$ is the number of tokens. The NMI is a function of $A$, ranging from 0 to 1. We highlight two

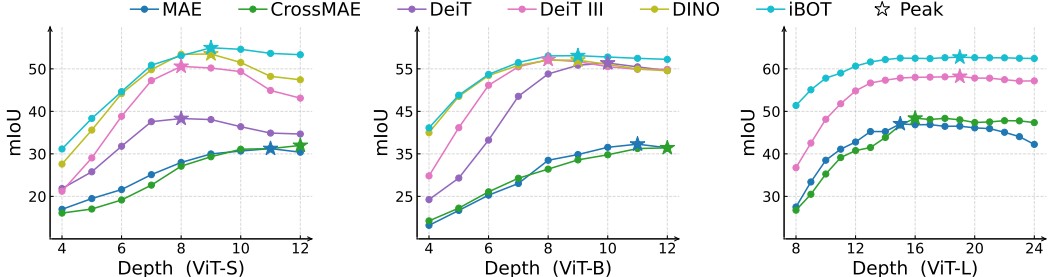

Figure 5: The layerwise linear probing performance of different methods on SODA (Li et al., 2021a).

special cases to clarify NMI: (1) When query tokens focus solely on their spatially corresponding key tokens (an extreme *local* pattern), $A$ becomes an identity matrix. Thus the joint probability of query and key tokens is equivalent to their marginal probability and the NMI reaches its maximum value of 1; (2) When all query tokens attend to the same key tokens (an extreme *global* pattern), each row of $A$ is identical. Consequently, the probability distribution of query and key tokens are independent, leading to the minimum NMI of 0.

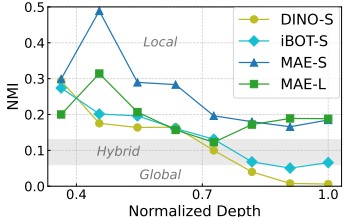

Figure 4: NMI on ImageNet.

**Therefore, *local* patterns have larger NMI values, while *global* patterns exhibit lower ones.** As illustrated in Fig. 4, the NMI of DINO-S and iBOT-S decreases with depth and consistently stays below that of MAE models. Especially, the NMI of DINO-S approaches 0 in later layers, revealing its *attention collapse*. In contrast, the NMI of MAE-L first decreases and then increases, due to the *hybrid* patterns in middle layers and *local* patterns in later layers.

### 3.3 THE HYBRID PATTERN MATTERS FOR SEMANTIC SEGMENTATION

Semantic segmentation is a dense prediction task where all pixels in an image are classified into different semantic classes. Local information is crucial as nearby pixels usually belong to the same class, while global cues are also essential since instances of the same class may appear in different positions within the image. Therefore, we hypothesize that **hybrid patterns, which capture both local and global information, are more important for semantic segmentation than purely *local* or *global* patterns**. To demonstrate this, we conduct the *layerwise linear probing* (LLP) experiments, where frozen features of only one layer are passed to the linear head, as shown in Fig. 8d.

**The LLP performance peaks where *hybrid* attention patterns emerge.** As shown in Fig. 5, supervised and CL methods peak at about three-quarters of the model's depth. Large MIM models (ViT-L) perform better in the middle layers. These peaks commonly occur near the *hybrid* patterns. In contrast, the performance of small MIM models (ViT-S and ViT-B) gradually increases with depth, peaking in the last two layers. This is because, although all layers exhibit *local* patterns, deep layers focus more on foreground tokens (Fig. 3b) and have smaller NMI values (Fig. 4), leading to better LLP performance. Additionally, the performance degradation in iBOT's deep layers is less pronounced than that in DINO and supervised methods. This aligns with observations that iBOT's deep-layer attention maps contain more local information (Sec. 3.1) and have larger NMI values than those of DINO (Fig. 4), underscoring the importance of the *hybrid* attention pattern.

**This hypothesis can explain the phenomena in Sec. 2.** Small MIM models struggle to learn the *hybrid* pattern, resulting in a notable performance gap compared to supervised and CL methods. Conversely, large MIM models successfully develop the *hybrid* pattern, making their fine-tuning performance comparable to other methods. iBOT performs best across different model sizes and evaluation metrics because the *hybrid* pattern occurs more frequently than in other methods.

### 3.4 THE TEXTURE BIAS HINDERS THE MODEL'S GENERALIZATION ON INFRARED IMAGES

When transferring RGB pre-trained models to infrared tasks, the distribution shift between these modalities significantly impacts performance. A major difference between RGB and infrared images is that RGB images can capture fine-grained textures, which are scarce in infrared images.

Therefore, we assume that **the model's bias towards texture would exacerbate the distribution shift, thereby impairing the transfer performance on infrared tasks.**

According to Park et al. (2023), MIM methods are texture-biased while CL and supervised methods are shape-biased. This bias is evident in attention maps in Fig. 3, where MAE models focus on textures while DINO emphasizes edges. Tab. 2 investigates the bias's impact on infrared segmentation. MAE-B outperforms DeiT-B and DINO-B on RGB datasets like

Table 2: The FT performance on RGB and infrared semantic segmentation datasets.

| Methods | RGB | | Infrared | | |
|---|---|---|---|---|---|
| | ADE20K | MFNet-RGB | MFNet-T | SODA | SCUT-Seg |
| DeiT-B | 47.4 | 57.07 | **48.59** | 69.73 | 69.35 |
| DINO-B | 46.8 | 55.20 | 48.54 | **69.79** | **69.82** |
| MAE-B | **48.1** | **57.29** | 46.78 | 68.18 | 67.86 |

ADE20K (Zhou et al., 2017) and MFNet-RGB (Ha et al., 2017), but consistently underperforms on infrared datasets. Notably, the paired MFNet-RGB and MFNet-T share the same scenario and image counts, differing only in modality. This indicates that MAE models pre-trained on ImageNet rely on low-level texture information to reconstruct masked patches, leading to poor generalization on texture-less infrared images. Therefore, **reducing the texture bias is a promising way to enhance the transfer performance of RGB-pre-trained models on infrared tasks.**

## 4 HOW TO IMPROVE THE PERFORMANCE ON INFRARED SEGMENTATION?

As discussed in Sec. 2.2, scaling up model sizes for better performance is impractical for resource-constrained scenarios. Therefore, we focus on enhancing small pre-trained models by introducing a comprehensive framework, UNIP, and validating its effectiveness through extensive experiments.

### 4.1 UNIP: A UNIFIED INFRARED PRE-TRAINING FRAMEWORK

UNIP improves small pre-trained models by optimizing the pre-training task, constructing an appropriate pre-training dataset, and refining the fine-tuning architecture, as depicted in Fig. 1.

**NMI-Guided Hybrid Attention Pattern Distillation (NMI-HAD).** Compressing knowledge from large models into smaller ones is an effective strategy to enhance performance without increasing parameter count. Previous works use various distillation targets like logits (Caron et al., 2021) and features (Xiong et al., 2024). However, they often overlook the relationship between distillation targets and attention patterns. As revealed in Sec. 3.3, the *hybrid* attention pattern is crucial for semantic segmentation, with NMI values linked to attention patterns. Therefore, we propose using *hybird* patterns as the distillation target and introduce the NMI-guided *hybrid* attention pattern distillation. First, the NMI value $\text{NMI}(A_l)$ of each teacher model's layer is calculated on ImageNet-1K:

$$\text{NMI}(A_l) = \frac{1}{M} \sum_{m=1}^{M} \text{NMI}(A_l^m), \quad A_l^m = \text{softmax}\left(\frac{Q_l^m(K_l^m)^T}{\sqrt{d}}\right), \quad l = \frac{L}{2} + 1, ..., L, \quad (1)$$

where $A_l^m$ denotes the $m$-th head attention matrix in the $l$-th layer. $L$ and $d$ are the number of layers and the dimension of the teacher model. The method for calculating NMI is detailed in Appendix D.1. Note that we only consider layers in the latter half of the model, as shallow layers do not capture sufficient knowledge. Next, NMI values are used to identify the location of the *hybrid* attention pattern. The attention map of the layer whose NMI is closest to an empirical value $s$ is utilized as the distillation target $A_T$. Finally, the attention map $A_S$ in the last layer of the student model is forced to imitate $A_T$ by employing the Kullback-Leibler (KL) divergence constraints:

$$A_T = \underset{A_l}{\arg\max} \, \Delta\text{NMI}(A_l), \quad \Delta\text{NMI}(A_l) = -|\text{NMI}(A_l) - s|, \quad \mathcal{L} = \frac{1}{M} \sum_{m=1}^{M} \text{KL}(A_T^m || A_S^m),$$
$$(2)$$

Empirically, the NMI values of *hybrid* patterns range between 0.06 and 0.12. We find that setting $s$ within this range yields good results. In all our experiments, we set it to 0.09 by default.

**InfMix Dataset.** To alleviate the distribution shift and reduce texture bias when distilling RGB pre-trained models for infrared tasks, we develop InfMix, a mixed dataset for distillation. InfMix comprises **859,375** images from both RGB and infrared modalities, constructed through four steps. (1) Infrared images play a key role in mitigating the distribution shift. However, existing datasets

Table 4: The infrared semantic segmentation performance of different models. Across various pre-trained methods, UNIP models significantly surpass pre-trained models of the same size. Remarkably, they even outperform their teacher models, despite the latter having more parameters.

| Methods | Params(M) | Fine-tuning (FT) | | | | Linear Probing (LP) | | | |
|---|---|---|---|---|---|---|---|---|---|
| | | SODA | MFNet-T | SCUT-Seg | Average FT | SODA | MFNet-T | SCUT-Seg | Average LP |
| MAE-L (Teacher) | 441.3 | 71.04 | 51.17 | 70.83 | 64.35 | 52.20 | 31.21 | 43.71 | 42.37 |
| MAE-T | 11.0 | 52.85 | 35.93 | 51.31 | 46.70 | 23.75 | 15.79 | 27.18 | 22.24 |
| UNIP-T | 11.0 | **64.83** | **48.77** | **67.22** | **60.27 (+13.57)** | **44.12** | **28.26** | **35.09** | **35.82 (+13.58)** |
| MAE-S | 41.9 | 63.36 | 42.44 | 60.38 | 55.39 | 38.17 | 21.14 | 34.15 | 31.15 |
| UNIP-S | 41.9 | **70.99** | **51.32** | **70.79** | **64.37 (+8.98)** | **55.25** | **33.49** | **43.37** | **44.04 (+12.89)** |
| MAE-B | 163.7 | 68.18 | 46.78 | 67.86 | 60.94 | 43.01 | 23.42 | 37.48 | 34.64 |
| UNIP-B | 163.7 | **71.47** | **52.55** | **71.82** | **65.28 (+4.34)** | **58.82** | **34.75** | **48.74** | **47.43 (+12.79)** |
| DINO-B (Teacher) | 163.7 | 69.79 | 48.54 | 69.82 | 62.72 | 59.33 | 34.86 | 47.23 | 47.14 |
| DINO-S | 41.9 | 68.56 | 47.98 | 68.74 | 61.76 | 56.02 | 32.94 | 45.94 | 44.97 |
| UNIP-S | 41.9 | **69.35** | **49.95** | **69.70** | **63.00 (+1.24)** | **57.76** | **34.15** | **46.37** | **46.09 (+1.12)** |
| iBOT-L (Teacher) | 441.3 | 71.75 | 51.66 | 71.49 | 64.97 | 61.73 | 36.68 | 50.12 | 49.51 |
| iBOT-S | 41.9 | 69.33 | 47.15 | 69.80 | 62.09 | 57.10 | 33.87 | 45.82 | 45.60 |
| UNIP-S | 41.9 | **70.75** | **51.81** | **71.55** | **64.70 (+2.61)** | **60.28** | **37.16** | **47.68** | **48.37 (+2.77)** |
| iBOT-B | 163.7 | 71.15 | 48.98 | 71.26 | 63.80 | 60.05 | 34.34 | 49.12 | 47.84 |
| UNIP-B | 163.7 | **71.75** | **51.46** | **72.00** | **65.07 (+1.27)** | **63.14** | **39.08** | **52.53** | **51.58 (+3.74)** |

often lack diversity and sufficient images, so we collect a large and unlabelled infrared pre-training dataset called **InfPre**. It consists of **541,088** images from **23** infrared-related datasets. Compared to the other two datasets in Tab. 3, InfPre offers a larger number of higher-resolution images sourced from more diverse datasets. Importantly, three segmentation datasets used in the benchmark are

Table 3: Comparisons of infrared pre-training datasets. #Subset denotes the number of datasets from which the images are collected.

| Dataset | #Image | #Subset | Width | Height |
|---|---|---|---|---|
| MSIP (Zhang et al., 2023) | 178,756 | 8 | 844 | 596 |
| Inf30 (Liu et al., 2024a) | 305,241 | - | 700 | 562 |
| InfPre (ours) | **541,088** | **23** | **1,075** | **686** |

excluded from InfPre for fair comparison. Details on data collection and deduplication can be found in Appendix F.1. (2) A subset of ImageNet-1K (Deng et al., 2009) is used, comprising **200,000** images evenly sampled from 1,000 classes. Since these images are part of the teacher model's pre-training data, they can anchor the student representation space close to the teacher's, thereby aiding in transferring the teacher's general feature extraction capabilities to the student. (3) The training set of COCO (Lin et al., 2014), with **118,287** images, is also included to further enrich the pre-training dataset. Unlike single-object-centric images in ImageNet, COCO images typically depict larger scenes with multiple objects, making them more similar to infrared images, as indicated in Tab. 21 in the appendix. (4) Images from ImageNet and COCO are converted to grayscale (three identical channels) to resemble infrared images more closely, as noted in Tab. 21.

**Last-Layer Feature Pyramid Network (LL-FPN).** To adapt the non-hierarchical ViT to multi-scale decoders in dense prediction tasks, previous works (He et al., 2022; Zhou et al., 2022) typically generate multi-scale feature maps from different layers of ViT, as shown in Fig. 8a. **However, we find this multi-layer design unnecessary for our distilled models.** In these models, the *hybrid* patterns in later layers equip the final features with both local and global information, making them suitable for multi-scale feature map generation. Inspired by ViTDet (Li et al., 2022b), we propose using the last-layer feature pyramid network during fine-tuning. It constructs all feature maps of different scales upon the last layer's features, as illustrated in Fig. 1 and Fig. 8c. As a bonus, this approach enhances the representation capacity of each scale branch compared to the configuration in Fig. 8a, since they go through the entire backbone, leading to improved fine-tuning performance.

## 4.2 EXPERIMENTS

The MAE-L, DINO-B, and iBOT-L are utilized as teacher models for distillation, and the 18th, 9th, and 21st layers are used as the target layer, according to Eq. (1) and Eq. (2). Unless otherwise specified, the distillation, fine-tuning, and linear probing processes are each conducted for 100 epochs. For ablation studies, we mainly focus on the fine-tuning metric as it reflects the model's highest achievable performance. More details about experimental settings can be found in Appendix C.4.

**Improvements of UNIP.** As shown in Tab. 4, UNIP significantly enhances the performance of small models across both metrics, often exhibiting comparable or even better performance than teacher

Table 5: Comparisons with other segmentation methods (FT). All the compared results except PAD are borrowed from TINN. Training epochs of SODA and MFNet-T are 200 and 300.

| Methods | Params(M) | SODA | MFNet-T |
|---|---|---|---|
| DeepLab V3+ (Chen et al., 2018) | 62.7 | 68.73 | 49.80 |
| PSPNet (Zhao et al., 2017) | 68.1 | 68.68 | 45.24 |
| UPerNet (Xiao et al., 2018) | 72.3 | 67.45 | 48.56 |
| SegFormer (Xie et al., 2021) | 84.7 | 67.86 | 50.68 |
| ViT-Adapter (Chen et al., 2023) | 99.8 | 68.12 | 50.62 |
| Mask2Former (Cheng et al., 2022) | 216.0 | 67.58 | 51.30 |
| MaskDINO (Li et al., 2023) | 223.0 | 66.32 | 51.03 |
| EC-CNN (Li et al., 2021a) | 54.5 | 65.87 | 47.56 |
| MCNet (Xiong et al., 2021) | 35.7 | 63.89 | 43.15 |
| PAD (MAE-B) (Zhang et al., 2023) | 164.9 | 69.71 | 50.14 |
| TINN (Chen & Bai, 2023) | 85.3 | 69.45 | 51.93 |
| UNIP-T (MAE-L) | 11.0 | 67.29 | 50.39 |
| UNIP-S (MAE-L) | 41.9 | 71.35 | 53.76 |
| UNIP-B (MAE-L) | 163.7 | 72.19 | 54.35 |

Table 6: Impact of distillation targets (UNIP-S).

| Target (Teacher) | Layer | SODA | MFNet-T | SCUT-Seg | Avg FT |
|---|---|---|---|---|---|
| Feature | 18 | 65.66 | 48.44 | 66.55 | 60.22 |
| (MAE-L) | 24 | 66.86 | 49.00 | 66.61 | 60.82 |
| Attention | 18 | 70.99 | 51.32 | 70.79 | 64.37 |
| (MAE-L) | 24 | 67.74 | 50.39 | 69.00 | 62.38 |

Table 7: Impact for the LL-FPN (UNIP-S). ✗ and ✓ denote the ones in Fig. 8a and Fig. 8c.

| Teacher (Layer) | LL-FPN | SODA | MFNet-T | SCUT-Seg | Avg FT |
|---|---|---|---|---|---|
| *Hybrid* Pattern | ✗ | 69.54 | 50.18 | 70.63 | 63.45 |
| MAE-L (Layer 18) | ✓ | 70.99 | 51.32 | 70.79 | 64.37 |
| *Local* Pattern | ✗ | 67.64 | 49.93 | 68.68 | 62.08 |
| MAE-L (Layer 24) | ✓ | 67.74 | 50.39 | 69.00 | 62.38 |
| *Hybrid* Pattern | ✗ | 68.56 | 49.44 | 68.49 | 62.16 |
| DINO-B (Layer 9) | ✓ | 69.35 | 49.95 | 69.70 | 63.00 |
| *Global* Pattern | ✗ | 68.62 | 47.36 | 69.71 | 61.90 |
| DINO-B (Layer 12) | ✓ | 68.50 | 48.40 | 69.67 | 62.19 |

models. With MAE-L as the teacher, UNIP-T, UNIP-S, and UNIP-B achieve average mIoU gains of **13.57%**, **8.98%**, and **4.34%** in fine-tuning, and **13.58%**, **12.98%**, and **12.79%** in linear probing. Notably, UNIP-S performs comparably to MAE-L with only **1/10** of the computational cost. UNIP-B even outperforms MAE-L by **0.93%** in FT and **5.06%** in LP. Using iBOT-L as the teacher, UNIP-S transcends iBOT-S by **2.61%** in FT and **2.77%** in LP. Meanwhile, UNIP-B shows gains of **1.27%** in FT and **3.74%** in LP, exceeding its teacher iBOT-L. Even with a smaller teacher like DINO-B, UNIP-S still enhances performance by at least **1.12%**. Tab. 5 compares the fine-tuning performance of UNIP with other RGB or infrared segmentation methods. With fewer than half the parameters, UNIP-S, distilled from MAE-L, surpasses the universal segmentation method Mask2Former (Cheng et al., 2022) by **3.77%** on SODA and **2.46%** on MFNet-T. It also outperforms TINN (Chen & Bai, 2023), specially designed for infrared semantic segmentation, by **1.9%** on SODA and **1.83%** on MFNet-T. A larger model UNIP-B further widens this performance gap, indicating that UNIP can greatly unleash the potential of the vanilla ViT for infrared semantic segmentation.

**Impact of Distillation Target Layers.** Fig. 6 displays the average fine-tuning performance using different layers of MAE-L and DINO-B as the distillation target layer. Notably, both models exhibit a strong positive correlation between average FT performance and ΔNMI in Eq. (2), as indicated by a large Pearson coefficient. Furthermore, the peaks of FT and ΔNMI occur in the same layer, highlighting the effectiveness of the NMI-HAD.

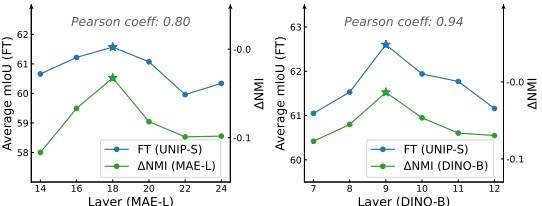

Figure 6: The average FT and NMI of each target layer. Each model is distilled for 20 epochs.

**Impact of the Hyperparameter** $s$**.** The parameter $s$ in Eq. (2) determines the layer chosen for distillation. As presented in Fig. 7, when $s$ ranges from 0.06 to 0.12, the selected layer remains nearly constant: the 18th layer for MAE-L and the 9th layer for DINO-B. Therefore, the performance of UNIP is relatively stable with respect to $s$.

**Comparison with the Feature Distillation.** As compared in Tab. 6, the performance of feature distillation consistently lags behind attention distillation across different layers, implying the latter's superiority. We believe this is because attention distillation only restricts the relationship between tokens, whereas feature distillation imposes direct constraints on each token's features. Excessive constraints on features may intensify the distribution shift and hinder the generalization of distilled models.

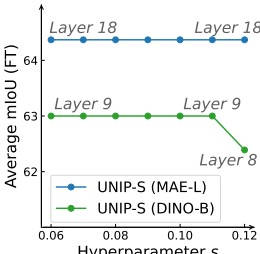

Figure 7: The average FT when employing different $s$ in Eq. (2).

Additionally, the performance of feature distillation across different layers is similar, likely due to the skip connections in ViT, which enhance feature similarities between layers. In contrast, the attention maps of different layers differ significantly, as revealed in Sec. 3.1.

**Impact of Pre-training Datasets.** Tab. 8 illustrates the performance of different datasets. As anticipated, all components of the InfMix dataset are necessary, including the infrared dataset InfPre, the ImageNet subset (Deng et al., 2009), the COCO training set (Lin et al., 2014), and the grayscale operation. Remarkably, InfMix significantly outperforms single-modality datasets like ImageNet and InfPre. This improvement can be attributed to the complementary strengths of both modalities:

Table 8: Ablations for components of the InfMix dataset. The teacher and student models are MAE-L and UNIP-S. All datasets are distilled for the same number of iterations for fair comparison.

| Dataset | #Images | SODA | MFNet-T | SCUT-Seg | Avg FT |
|---|---|---|---|---|---|
| InfMix | 859,375 | **70.99** | **51.32** | 70.79 | **64.37** |
| – w/o IN1K | 659,375 | 69.41 | 51.13 | 70.21 | 63.58 |
| – w/o COCO | 741,088 | 69.62 | 51.29 | 69.58 | 63.50 |
| – w/o Grayscale | 859,375 | 69.73 | 50.71 | **71.09** | 63.84 |
| ImageNet-1K | 1,281,167 | 69.39 | 49.11 | 69.63 | 62.71 |
| InfPre | 541,088 | 68.45 | 51.27 | 67.87 | 62.53 |

infrared images help mitigate the distribution shift issue, while RGB images enhance general feature extraction capabilities. The mixed dataset effectively balances these two aspects. Moreover, Tab. 15 in the appendix displays the scaling characteristics of pre-training data, demonstrating the necessity of constructing the larger InfMix dataset.

**Comparison with Contiual Pre-training on Target Domain.** We initialize MAE-S with RGB pre-trained weights and further pre-train it on InfMix for 100 epochs. As shown in Tab. 9, this continually pre-trained MAE-S (58.53%) exceeds

Table 9: Comparisons of pre-training methods.

| Method | Avg FT | Training Time (h) |
|---|---|---|
| Continual Pre-trained (MAE-S) | 58.53 | 75.0 (1x RTX3090) |
| UNIP-S (MAE-L distilled) | **64.37** | **72.5** (1x RTX3090) |

the RGB pre-trained one (55.39% in Tab. 4). However, it still underperforms UNIP-S by 5.84% and requires more training time, highlighting the efficiency of UNIP over continual pre-training.

**Impact of the LL-FPN.** Tab. 7 shows the performance of models distilled from various teachers. While LL-FPN enhances performance for all models, the improvements are much greater when using *hybrid* patterns as distillation targets than *local* or *global* patterns. This demonstrates LL-FPN's superiority and good compatibility with the *hybrid* pattern, supporting the analysis in Sec. 4.1.

**Applicability to Other Modalities.** We extend the LLP experiments in Sec. 3.3 to the RGB and depth modalities. As shown in Tab. 10, for both DINO-S and DeiT-S, the LLP performance of middle layers (the *local* pattern) surpasses that of deep layers (the *global* pattern) across all RGB and depth semantic segmenta-

Table 10: The LLP performance on RGB and depth datasets. Training epochs are 30 for ADE20K and 100 for others.

| (Modality) Dataset | DINO-S | | DeiT-S | |
|---|---|---|---|---|
| | Layer 9 (*Hybrid*) | Layer 12 (*Global*) | Layer 9 (*Hybrid*) | Layer 12 (*Global*) |
| (RGB) ADE20K (Zhou et al., 2017) | **26.11** | 23.15 | **24.35** | 22.68 |
| (RGB) MFNet-RGB (Ha et al., 2017) | **38.94** | 37.53 | **30.43** | 29.44 |
| (Depth) NYUDepthv2 (Silberman et al., 2012) | **17.25** | 15.29 | **5.55** | 5.15 |
| (Depth) SUN-RGBD (Song et al., 2015) | **13.17** | 11.41 | **5.61** | 4.94 |

tion datasets. This mirrors the phenomenon in the infrared domain discussed in Sec. 3.3, underscoring the importance of *hybrid* patterns for semantic segmentation tasks, regardless of dataset size or modality. Therefore, we believe that UNIP can be effectively extended to other modalities.

## 5    CONCLUSION AND DISCUSSION

In this work, we comprehensively benchmark the infrared segmentation performance of different pre-training methods and uncover several valuable insights. We further analyze the pre-trained attention maps and identify the importance of *hybrid* patterns for semantic segmentation. Finally, we propose the UNIP framework to improve the performance of small ViT models. Extensive experimental results demonstrate the effectiveness of our dataset and method. UNIP presents a viable approach for selective knowledge distillation in domain transfer settings. We hope our analysis can provide meaningful insights into the characteristics and differences among pre-training methods, ultimately contributing to the advancements of visual pre-training and downstream transfer learning.

**Limitations and Future Work.** Due to limited computing resources, we validate UNIP's effectiveness only in the infrared domain for semantic segmentation. However, we believe UNIP can be effectively extended to other modalities, such as RGB and depth images, as the superiority of *hybrid* patterns in these modalities is demonstrated in Tab. 10. Exploring its potential in other dense prediction tasks, like object detection and depth estimation, is also worthwhile. Moreover, combining *hybrid* patterns from different pre-trained methods could be a promising avenue.

## ACKNOWLEDGMENTS

This research was supported by the Strategic Priority Research Program of the Chinese Academy of Sciences, Grant No.XDA0370305, the Strategic Priority Research Program of the Chinese Academy of Sciences, Grant No.XDB0500103, and the National Natural Science Foundation of China under Grant 62306310.

## REPRODUCIBILITY STATEMENT

Reproducibility is a priority in our research. In this statement, we outline the measures taken to ensure our work can be reproduced.

**Source Code.** The source code of our work is available at this link. Researchers can access and utilize our code to reproduce the experimental results in this paper. The source code and pre-trained model weights will be made publicly available.

**Experimental Setup and Details.** In the main paper, the basic experimental configurations are presented in Sec. 2.1 (benchmark) and in Sec. 4.2 (UNIP). In Appendix C, we provide the detailed settings, including the implementation details of the benchmark (Appendix C.1) and UNIP (Appendix C.4), the comparisons of different evaluation metrics and their hyperparameter settings (Appendix C.2), and the evaluation datasets usage (Appendix C.3).

**Datasets.** We outline the construction steps of our InfMix dataset in Sec. 4.1. In Appendix F, we further present more details about the dataset collection and preprocessing.

By highlighting these references, we intend to improve the reproducibility of our work, helping other researchers verify and build on our findings. We're open to any questions or requests for more information about our methods, as we aspire to ensure our research is transparent and reliable.

## ETHICS STATEMENT

Our constructed dataset InfMix is derived from the extraction and integration of 25 publicly accessible RGB and infrared datasets. These datasets are available for public use. During the extraction process, we employ data deduplication methods only, ensuring that no personal biases were introduced. We adhere to all usage requirements of the original datasets and use them solely for scientific research purposes. Importantly, the dataset does not include any additional metadata or labels that could be used to identify individuals. Detailed information regarding the composition and construction process of our dataset is provided in Appendix F. We are solely responsible for any legal violations with respect to our collected dataset and accept all the risks associated.

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

APPENDIX

In Sec. A, we discuss the related works. In Sec. B, we review the relationship between our motivations and proposed methods. In Sec. C, we provide detailed descriptions of the experimental settings, including the complete benchmark results, evaluation metrics and datasets, and the experimental specifics of UNIP. Further analysis is conducted in Sec. D, covering (1) the relationship between NMI and attention patterns, and (2) the CKA analysis of feature representation. Additional experimental results are presented in Sec. E. Finally, we provide more details of the pre-training dataset in Sec. F and offer additional visualization results in Sec. G.

## A    RELATED WORK

**Visual pre-training** aims to equip models with fundamental feature extraction capabilities using large-scale pre-training data, aiding their fine-tuning on downstream tasks. Supervised pre-training (He et al., 2016; Dosovitskiy et al., 2021), one of the earliest methods, typically involves image classification on labeled datasets like ImageNet (Deng et al., 2009). However, its reliance on labeled data limits its scalability, prompting the development of self-supervised pre-training. This approach utilizes various pretext tasks, such as contrastive learning and masked image modeling, to pre-train models, achieving results competitive with supervised counterparts. These methods are detailed in Sec. 2.1. In the infrared domain, Zhang et al. (2023) proposes the patchwise-scale adapter to adapt RGB pre-trained models for infrared tasks, and Liu et al. (2024a) constructs a hierarchical model for infrared pre-training. However, previous works have not thoroughly analyzed the transfer performance of different pre-training methods on infrared tasks. Our work aims to fill this gap.

**Knowledge distillation (KD)** is a widely used technique to improve the performance of small models by extracting knowledge from well-trained large models. Initially developed for supervised learning (Hinton et al., 2015), it has recently gained popularity in self-supervised learning. Bai et al. (2023), Liu et al. (2024b), and Xiong et al. (2024) focus on feature KD, while Caron et al. (2021), Zhou et al. (2022), and Oquab et al. (2024) employ self-relational KD. Similar to our work, Wang et al. (2023) and Ren et al. (2023) explore attention KD, but they only conduct empirical explorations on MAE in the RGB domain and do not explore the underlying mechanism of using different layers for distillation. In contrast, our research systematically investigates which attention patterns are most advantageous for distillation in domain transfer settings and proposes the NMI metric to guide the process, demonstrating effectiveness across various pre-training methods.

**Semantic segmentation** is a widely investigated visual task that aims to classify each pixel into different semantic categories. As one of the fundamental works, FCN (Long et al., 2015) employs a fully convolutional neural network for pixel-to-pixel classification. The following works (Chen et al., 2018; Zhao et al., 2017; Xiao et al., 2018) enhance FCN by constructing the feature pyramid network and improving the context fusion module. With the advancements of transformer-based architectures in visual tasks, Xie et al. (2021) proposes the powerful SegFormer, featuring a hierarchical transformer encoder and a lightweight decoder. Mask2Former (Cheng et al., 2022) further unifies semantic segmentation with other segmentation tasks following the framework of DETR (Carion et al., 2020). For infrared semantic segmentation, Xiong et al. (2021) develops a multi-level correction network (MCNet) to capture the context in infrared images, while TINN (Chen & Bai, 2023) focuses on preserving the inherent radiation characteristic within the thermal imaging process. However, these methods do not explore the impact of different pre-trained models on segmentation performance. Our study utilizes semantic segmentation as a representative downstream visual task and systematically investigates the influence of various pre-trained models on this task.

## B    RELATIONSHIPS BETWEEN MOTIVATIONS AND METHODS

Our primary motivation is to enhance the performance of models on infrared semantic segmentation tasks. From the model perspective, factors affecting the performance on specific tasks include not only the design of the model architecture but also the quality of the model's pre-training. Previous works (Li et al., 2021a; Chen & Bai, 2023) have aimed to improve performance by designing specific network architectures for infrared semantic segmentation tasks. However, in the infrared domain, where labeled data is limited, the quality of the pre-trained model is also crucial. **Therefore, our**

**work explores an alternative approach by emphasizing the optimization of pre-trained models specifically for infrared semantic segmentation tasks to enhance performance.** To facilitate this exploration, our work is organized into three stages: benchmark establishment, cause analysis, and method proposal, as illustrated in Fig. 1.

**Benchmark Establishment (Sec. 2).** We establish a benchmark for the transfer performance of six RGB pre-training methods, encompassing a total of 18 pre-trained models, on three infrared semantic segmentation datasets (Sec. 2.1). Our findings reveal several key phenomena (Sec. 2.2), such as the lack of correlation between model performance on ImageNet and infrared segmentation datasets (Tab. 1), and the superior generalization of supervised and contrastive learning methods over masked image modeling methods in the context of infrared segmentation tasks (Fig. 2).

**Cause Analysis (Sec. 3).** To analyze the performance discrepancies among various pre-training methods in infrared segmentation tasks, we conduct an in-depth analysis of the attention maps from the pre-trained models. Our findings indicate that the degree of focus on local and global information (Sec. 3.1 - Sec. 3.3), as well as on shape and texture information (Sec. 3.4), significantly impacts the performance of infrared semantic segmentation tasks. We further validate through corresponding experiments that the existence of hybrid attention patterns (Fig. 3 - Fig. 5) and the reduced bias towards texture (Tab. 2) both play crucial roles in enhancing the performance of pre-trained models in infrared segmentation tasks.

**Method Proposal (Sec. 4.1).** Based on the observations and analyses from the previous two sections, we propose UNIP, a framework designed to improve the infrared segmentation performance of pre-trained models through three key aspects: the pre-training objective (NMI-HAD), the pre-training data (InfMix), and the fine-tuning architecture (LL-FPN). Both NMI-HAD and LL-FPN enhance performance by effectively leveraging hybrid attention patterns, while InfMix enhances performance by reducing the pre-trained model's bias toward texture information. **Importantly, our approach does not alter the structure of the backbone model or the decoder; instead, we focus on targeted pre-training specifically designed for infrared segmentation tasks. We believe this is one way in which the proposed method is specific to infrared semantic segmentation tasks.** As a result, our pre-trained models significantly outperform RGB pre-trained models of comparable or even larger sizes in infrared semantic segmentation tasks (Tab. 4), and also achieve superior performance compared to other models specifically designed for infrared segmentation (Tab. 5).

# C  EXPERIMENTAL DETAILS

## C.1  BENCHMARK DETAILS

**Reproduction of small MAE models.** The MAE-T and MAE-S are reproduced following the settings in He et al. (2022). We make several adjustments to the decoder to make it suitable for small encoders. For both MAE-S and MAE-T, the decoder includes 8 transformer blocks, each with 8 attention heads. The decoder dimensions in MAE-S and MAE-T are 256 and 192, respectively.

**Implementation details.** The weights of all pre-trained models are downloaded from corresponding official repositories. The models are trained for 100 epochs using MMSegmentation (Contributors, 2020). For different methods and model sizes, we keep the learning rate constant and sweep the layerwise decay rate across {0.5, 0.65, 0.75, 0.85, 1.0}. To adapt models pre-trained on three-channel RGB images for single-channel infrared images, we duplicate the infrared images three times to create pseudo-three-channel images.

## C.2  EVALUATION METRICS

**Fine-tuning.** *Fine-tuning* is the default evaluation metric in this work, which utilizes the pre-trained model as the backbone of existing semantic segmentation models. Following previous works (He et al., 2022; Zhou et al., 2022), we employ UperNet (Xiao et al., 2018) as the semantic segmentation model. As illustrated in Fig. 8a, to build the feature pyramid based on the non-hierarchical ViT model, features from different layers are passed through the MaxPooling layers or DeConv layers, to obtain features of different resolutions. These multi-scale features are then input into the decoder for segmentation results. Following He et al. (2022) and Zhou et al. (2022), we use features of the {4, 6, 8, 12} layers in ViT-T, ViT-S, and ViT-B, and the features of the {8, 12, 16, 24} layers in

Table 11: The performance of different pre-trained models on ImageNet and infrared semantic segmentation datasets. The *Scratch* means the performance of randomly initialized models. The *PT Epochs* denotes the pre-training epochs while the *IN1K FT epochs* represents the fine-tuning epochs on ImageNet (Deng et al., 2009). † denotes models reproduced using official codes. ⋆ refers to the effective epochs used in Zhou et al. (2022). The top two results are marked in **bold** and underlined format. Supervised and CL methods, MIM methods, and UNIP models are colored in orange, gray, and cyan, respectively.

| Methods | PT Epochs | IN1K FT Epochs | IN1K FT Acc | Fine-tuning (FT) SODA | MFNet-T | SCUT-Seg | Mean | Linear Probing (LP) SODA | MFNet-T | SCUT-Seg | Mean |
|---|---|---|---|---|---|---|---|---|---|---|---|
| ViT-Tiny/16 | | | | | | | | | | | |
| Scratch | - | - | - | 31.34 | 19.50 | 41.09 | 30.64 | - | - | - | - |
| MAE† (He et al., 2022) | 800 | 200 | 71.8 | 52.85 | 35.93 | 51.31 | 46.70 | 23.75 | 15.79 | 27.18 | 22.24 |
| DeiT (Touvron et al., 2021) | 300 | - | 72.2 | 63.14 | 44.60 | 61.36 | 56.37 | 42.29 | 21.78 | 31.96 | 32.01 |
| UNIP (MAE-L) | 100 | - | - | 64.83 | 48.77 | 67.22 | 60.27 | 44.12 | 28.26 | 35.09 | 35.82 |
| UNIP (iBOT-L) | 100 | - | - | 65.54 | 48.45 | 67.73 | 60.57 | 52.95 | 30.10 | 40.12 | 41.06 |
| ViT-Small/16 | | | | | | | | | | | |
| Scratch | - | - | - | 41.70 | 22.49 | 46.28 | 36.82 | - | - | - | - |
| MAE† (He et al., 2022) | 800 | 200 | 80.0 | 63.36 | 42.44 | 60.38 | 55.39 | 38.17 | 21.14 | 34.15 | 31.15 |
| CrossMAE (Fu et al., 2024) | 800 | 200 | 80.5 | 63.95 | 43.99 | 63.53 | 57.16 | 39.40 | 23.87 | 34.01 | 32.43 |
| DeiT (Touvron et al., 2021) | 300 | - | 79.9 | 68.08 | 45.91 | 66.17 | 60.05 | 44.88 | 28.53 | 38.92 | 37.44 |
| DeiT III (Touvron et al., 2022) | 800 | - | 81.4 | 69.35 | 47.73 | 67.32 | 61.47 | 54.17 | 32.01 | 43.54 | 43.24 |
| DINO (Caron et al., 2021) | 3200⋆ | 200 | 82.0 | 68.56 | 47.98 | 68.74 | 61.76 | 56.02 | 32.94 | 45.94 | 44.97 |
| iBOT (Zhou et al., 2022) | 3200⋆ | 200 | 82.3 | 69.33 | 47.15 | 69.80 | 62.09 | 57.10 | 33.87 | 45.82 | 45.60 |
| UNIP (DINO-B) | 100 | - | - | 69.35 | 49.95 | 69.70 | 63.00 | 57.76 | 34.15 | 46.37 | 46.09 |
| UNIP (MAE-L) | 100 | - | - | 70.99 | 51.32 | 70.79 | 64.37 | 55.25 | 33.49 | 43.37 | 44.04 |
| UNIP (iBOT-L) | 100 | - | - | 70.75 | 51.81 | 71.55 | 64.70 | 60.28 | 37.16 | 47.68 | 48.37 |
| ViT-Base/16 | | | | | | | | | | | |
| Scratch | - | - | - | 44.25 | 23.72 | 49.44 | 39.14 | - | - | - | - |
| MAE (He et al., 2022) | 1600 | 100 | 83.6 | 68.18 | 46.78 | 67.86 | 60.94 | 43.01 | 23.42 | 37.48 | 34.64 |
| CrossMAE (Fu et al., 2024) | 800 | 100 | 83.7 | 68.29 | 47.85 | 68.39 | 61.51 | 43.35 | 26.03 | 38.36 | 35.91 |
| DeiT (Touvron et al., 2021) | 300 | - | 81.8 | 69.73 | 48.59 | 69.35 | 62.56 | 57.40 | 34.82 | 46.44 | 46.22 |
| DeiT III (Touvron et al., 2022) | 800 | 20 | 83.8 | 71.09 | 49.62 | 70.19 | 63.63 | 59.01 | 35.34 | 48.01 | 47.45 |
| DINO (Caron et al., 2021) | 1600⋆ | 100 | 83.6 | 69.79 | 48.54 | 69.82 | 62.72 | 59.33 | 34.86 | 47.23 | 47.14 |
| iBOT (Zhou et al., 2022) | 1600⋆ | 100 | 84.0 | 71.15 | 48.98 | 71.26 | 63.80 | 60.05 | 34.34 | 49.12 | 47.84 |
| UNIP (MAE-L) | 100 | - | - | 71.47 | 52.55 | 71.82 | 65.28 | 58.82 | 34.75 | 48.74 | 47.43 |
| UNIP (iBOT-L) | 100 | - | - | 71.75 | 51.46 | 72.00 | 65.07 | 63.14 | 39.08 | 52.53 | 51.58 |
| ViT-Large/16 | | | | | | | | | | | |
| Scratch | - | - | - | 44.70 | 23.68 | 49.55 | 39.31 | - | - | - | - |
| MAE (He et al., 2022) | 1600 | 50 | 85.9 | 71.04 | 51.17 | 70.83 | 64.35 | 52.20 | 31.21 | 43.71 | 42.37 |
| CrossMAE (Fu et al., 2024) | 800 | 50 | 85.4 | 70.48 | 50.97 | 70.24 | 63.90 | 53.29 | 33.09 | 45.01 | 43.80 |
| DeiT3 (Touvron et al., 2022) | 800 | 20 | 84.9 | 71.67 | 50.78 | 71.54 | 64.66 | 59.42 | 37.57 | 50.27 | 49.09 |
| iBOT (Zhou et al., 2022) | 1000⋆ | 50 | 84.8 | 71.75 | 51.66 | 71.49 | 64.97 | 61.73 | 36.68 | 50.12 | 49.51 |

ViT-L, to build the feature pyramid. Remarkably, in fine-tuning, all parameters including the pre-trained model, the feature pyramid, and the decoder, are tuned with the labeled downstream datasets. Hyperparameters are listed in Tab. 12.

**Linear Probing.** As mentioned above, *fine-tuning* introduces additional learnable parameters and alters the pre-trained feature representation. Its performance may not fully reflect the characteristics of the pre-trained features. Therefore, *linear probing* is also employed as an evaluation metric. As shown in Fig. 8b, features from different layers are resized to $1/4$ of the input resolution and then concatenated together. Finally, a linear head ($1 \times 1$ conv) utilizes these concatenated features to predict segmentation results. Notably, only the linear head is trainable, while all other parameters are frozen. The layer settings of output features are the same as *fine-tuning*.

**Fine-tuning (LL-FPN).** This metric is discussed in Sec. 4, which aims to enhance the fine-tuning performance of UNIP models by using the last layer to obtain features of different resolutions, as depicted in Fig. 8c. Specifically, we employ the features of the $\{12, 12, 12, 12\}$ layers in ViT-T, ViT-S, and ViT-B, and the features of the $\{24, 24, 24, 24\}$ layers in ViT-L, to build the feature pyramid. Other settings remain the same as *fine-tuning*.

**Layerwise Linear Probing.** This metric is a layerwise version of the *linear probing* metric. It is designed to assess the pre-trained feature representation at each layer. As shown in Fig. 8d, only the features of a single layer are forwarded to the linear head following the resize operation. Other settings are the same as *linear probing*.

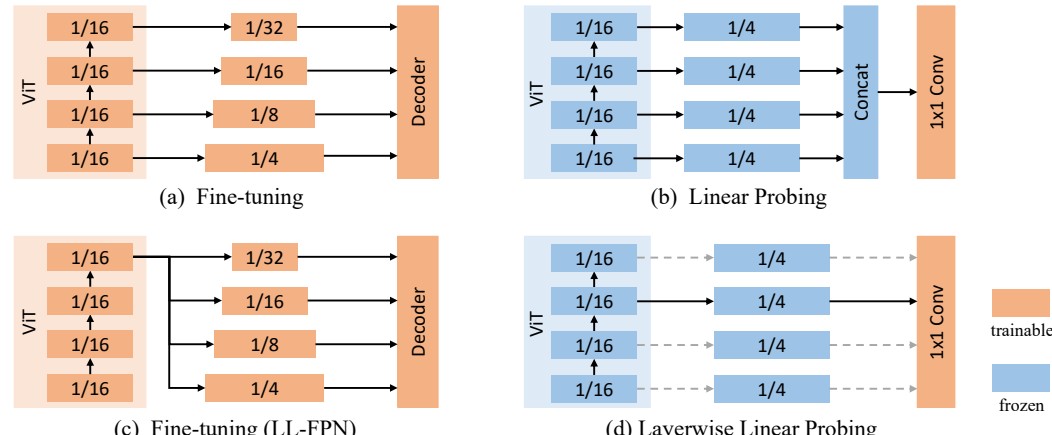

Figure 8: Illustrations of different transfer architectures for semantic segmentation tasks.

Table 12: Settings of semantic segmentation.

| Hyperparameters | SODA | MFNet-T | SCUT-seg |
|---|---|---|---|
| Input resolution | $512 \times 512$ | $512 \times 512$ | $512 \times 512$ |
| Training epochs | 100 / 200 | 100 / 300 | 100 |
| Training iterations | 14400 / 28800 | 9800 / 29400 | 16800 |
| Peak learning rate | 1e-4 | 1e-4 | 1e-4 |
| Batch size | 8 | 8 | 8 |
| Optimizer | AdamW | AdamW | AdamW |
| Weight decay | 0.05 | 0.05 | 0.05 |
| Optimizer momentum | $\beta_1, \beta_2 = 0.9, 0.999$ | $\beta_1, \beta_2 = 0.9, 0.999$ | $\beta_1, \beta_2 = 0.9, 0.999$ |
| Learning rate schedule | Linear decay | Linear decay | Linear decay |
| Minimal learning rate | 0 | 0 | 0 |
| Warmup steps | 1500 / 3000 | 1000 / 3000 | 1700 |

## C.3 EVALUATION DATASETS

**SODA** (Li et al., 2021a). This dataset features a variety of indoor and outdoor scenes. It comprises 1,168 training images and 1,000 test images, spanning 20 distinct semantic categories, including road, building, car, chair, lamp, table, monitor, and others.

**MFNet** (Ha et al., 2017). This dataset focuses on RGBT semantic segmentation for automotive driving scenarios and includes 1,569 image pairs of infrared and RGB images. It is divided into 784 training images, 392 validation images, and 393 test images, covering 8 semantic categories such as car, person, bike, curve, and others. When benchmarking the performance of different pre-training methods, we combine the validation set with the test set, resulting in a larger test set of 785 images. When comparing UNIP models with other SOTA semantic segmentation models, we follow their settings, *i.e.*, using the original 393 test images for evaluation.

**SCUT-Seg** (Xiong et al., 2021). This dataset includes 1345 training images and 665 test images in nighttime driving scenes. It has 10 classes including road, person, fence, pole, and others.

**ADE20K** (Zhou et al., 2017). ADE20K is a large-scale RGB semantic segmentation dataset, covering a variety of scenes from indoor to outdoor and nature to urban. It consists of 20,210 training images and 2,000 test images, with 150 different semantic categories.

**ImageNet-1K** (Deng et al., 2009). ImageNet-1K is a subset of the ImageNet database, consisting of 1,000 categories with roughly 1.2 million training images, 50,000 validation images, and 100,000 test images. It is widely used in computer vision research like image classification and pre-training.

## C.4 UNIP.

**Head Misalignment.** To solve the head misalignment between teacher and student models during distillation, we experiment with two methods. (1) The first method is the adaptive block proposed

Table 13: Configurations of ViT for semantic segmentation tasks.

| Model | Dimension | Head Num | Depth |
|-------|-----------|----------|-------|
| ViT-T | 192 | 3 | 12 |
| ViT-S | 384 | 6 | 12 |
| ViT-B | 768 | 12 | 12 |
| ViT-L | 1024 | 16 | 24 |

Table 14: Settings of pre-training.

| Hyperparameters | Value |
|-----------------|-------|
| Input resolution | $224 \times 224$ |
| Training epochs | 100 |
| Warmup epochs | 5 |
| Optimizer | AdamW |
| Base learning rate | 1e-4 |
| Weight decay | 0.05 |
| Optimizer momentum | $\beta_1, \beta_2 = 0.9, 0.95$ |
| Batch size | 4096 |
| Learning rate schedule | Cosine decay |
| Augmentation | Random resized cropping & Random horizontal flipping |

in Ren et al. (2023). Specifically, during distillation, the number of attention heads in the student model's last layer is adjusted to be the same as that of the teacher model by changing the head dimension while keeping the overall dimension constant. When performing fine-tuning or linear probing on downstream tasks, the number of attention heads is reverted to the standard setting in Tab. 13. (2) The second method involves adding a self-attention layer at the end of the student model during distillation. The number of attention heads in the extra attention layer is equivalent to the teacher model's. This layer is removed when transferring to downstream tasks. These two methods achieve similar performance, but the latter consumes slightly more training time. Therefore, we use the first method in practice.

**Feature Distillation.** For the feature distillation in Tab. 6, we employ a linear projection layer to match the dimension of the student model to that of the teacher model. The distillation and fine-tuning settings are the same as UNIP. The loss function is the cosine similarity loss between the $L_2$ normalized student feature $l_2(F_T)$ and teacher feature $l_2(F_S)$:

$$L = 1 - \cos(l_2(F_T) \cdot l_2(F_S)). \tag{3}$$

**Implementation Details.** All experiments are conducted using the PyTorch toolkit (Paszke et al., 2019) on 8 NVIDIA RTX 3090 GPUs. The default settings are shown in Tab. 14. We use the linear *learning rate* scaling rule: $lr = base\_lr \times$ batchsize / 256, following He et al. (2022). The semantic segmentation settings of UNIP models are the same as those in Appendix C.2.

# D ADDITIONAL ANALYSIS

## D.1 NORMALIZED MUTUAL INFORMATION

The Normalized Mutual Information (NMI) is employed in Sec. 3.2 to measure the attention patterns. Let $p(q_i)$ denote the marginal probability of the $i$-th query token and $p(k_j)$ denote the marginal probability of the $j$-th key token. Since query tokens are evenly distributed across every spatial coordinate, $p(q_i)$ can be formulated as:

$$p(q_i) = \frac{1}{N}, \quad i = 1, 2, ..., N. \tag{4}$$

Assume $A^m \in \mathbb{R}^{N \times N}$ represents the $m$-th head of the attention matrix after the softmax operation without the *class* token, where $N$ is the number of spatial tokens. The attention scores from each query token to all key tokens sum to 1, *i.e.*, $\sum_{j=1}^{N} A_{i,j}^m = 1, i = 1, 2, ..., N$. Thus, each row of $A$ can be viewed as the conditional probability distribution of key tokens given the query token:

$$p(k_j|q_i) = A_{i,j}^m. \tag{5}$$

Then the joint probability of $q_i$ and $k_j$ can be calculated as:

$$p(q_i, k_j) = p(k_j|q_i)p(q_i) = \frac{1}{N} A_{i,j}^m. \tag{6}$$

The marginal probability of $k_j$ is:

$$p(k_j) = \sum_{i=1}^{N} p(q_i, k_j) = \frac{1}{N} \sum_{i=1}^{N} A_{i,j}^m. \tag{7}$$

The mutual information of query and key tokens can be formulated as:

$$
\begin{aligned}
I^m(Q;K) &= \sum_{i=1}^{N}\sum_{j=1}^{N} p(q_i,k_j) \log \frac{p(q_i,k_j)}{p(q_i)p(k_j)} \\
&= \sum_{i=1}^{N}\sum_{j=1}^{N} \frac{1}{N} A_{i,j}^m \log \frac{N A_{i,j}^m}{\sum_{i=1}^{N} A_{i,j}^m}.
\end{aligned}
\tag{8}
$$

The entropy of query and key tokens can be calculated as:

$$
H^m(Q) = -\sum_{i=1}^{N} p(q_i) \log p(q_i) = -\sum_{i=1}^{N} \frac{1}{N} \log \frac{1}{N},
\tag{9}
$$

$$
H^m(K) = -\sum_{i=1}^{N} p(k_j) \log p(k_j) = -\sum_{j=1}^{N} \left( \frac{1}{N} \sum_{i=1}^{N} A_{i,j}^m \log \frac{1}{N} \sum_{i=1}^{N} A_{i,j}^m \right).
\tag{10}
$$

Therefore, the NMI of the $m$ head is:

$$
\mathrm{NMI}^m(Q;K) = \frac{I^m(Q;K)}{\sqrt{H^m(Q)H^m(K)}}.
\tag{11}
$$

The final NMI is calculated by averaging on all heads:

$$
\mathrm{NMI}(Q;K) = \frac{1}{M} \sum_{m=1}^{M} \mathrm{NMI}^m(Q;K).
\tag{12}
$$

The value of NMI ranges from 0 to 1. It reaches the maximum value of 1 when the joint probability of the query and key tokens is the same as their marginal probability:

$$
p(q_i,k_i) = p(q_i) = p(k_i), \quad i = 1,2,...,N.
\tag{13}
$$

According to Eq. (4), Eq. (6), Eq. (7), and Eq. (13), it can be derived that

$$
A_{i,j}^m =
\begin{cases}
1, & \text{if } i = j, \\
0, & \text{if } i \neq j,
\end{cases}
\tag{14}
$$

which implies that the attention matrix of each head is an identity matrix. This indicates that each query token focuses only on the key token at the same spatial position, which is a particular case of the *local* attention pattern.

On the other hand, the NMI has a value of 0 when the query and key tokens are independent:

$$
p(q_i,k_j) = p(q_i)p(k_j), \quad i = 1,2,...,N, j = 1,2,...,N.
\tag{15}
$$

According to Eq. (4), Eq. (6), Eq. (7), and Eq. (15), we can derive that

$$
A_{i,j}^m = A_{k,j}^m, \quad i = 1,2,...,N, k = 1,2,...,N, j = 1,2,...N,
\tag{16}
$$

which indicates that every row of the attention matrix is the same. This means that each query token has the same attention maps for all key tokens, which is a particular case of the *global* attention pattern. **Therefore, a higher NMI value indicates a stronger relationship between the query and key tokens and a more local attention pattern. Conversely, a lower NMI value means that different query tokens have more similar and global attention patterns for key tokens.**

### D.2 CENTERED KERNEL ALIGNMENT

In this section, we extend the analysis in Sec. 3.1 from the attention pattern to the feature representation. Let $x^l$ denote the input features of the $l$-th block of the ViT model. The features of the next block $x^{l+1}$ can be formulated as:

$$
x_{tmp} = x^l + \mathrm{Attention}(\mathrm{LN}(x^l),
\tag{17}
$$

$$
x^{l+1} = x_{tmp} + \mathrm{FFN}(\mathrm{LN}(x_{tmp}),
\tag{18}
$$

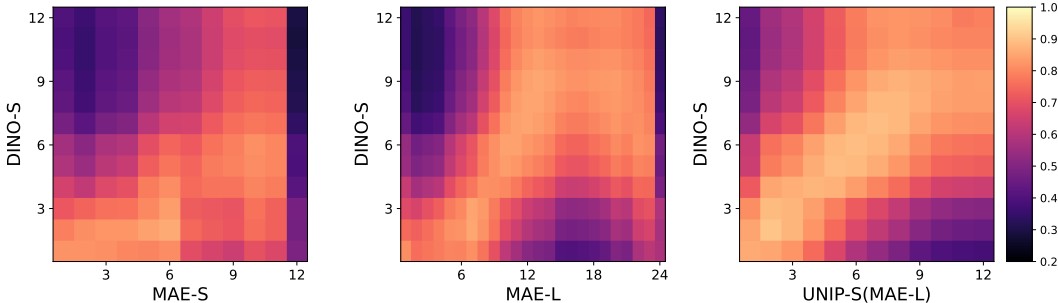

Figure 9: CKA representation analysis of different models. UNIP-S aligns well with DINO-S in the shallow and middle layers, indicating that the *hybrid* patterns are effectively distilled from MAE-L.

Table 15: The FT performance of using different ratios of InfMix as the pre-training dataset. The images are evenly sampled from each sub-dataset. The teacher and student models are MAE-L and UNIP-S.

| Ratio | #Images | SODA | MFNet-T | SCUT-Seg | Avg FT |
|---|---|---|---|---|---|
| 1% | 8,594 | 20.61 | 16.10 | 31.24 | 22.65 |
| 10% | 85,938 | 59.66 | 38.80 | 57.72 | 52.06 |
| 30% | 257,813 | 68.29 | 50.39 | 69.04 | 62.57 |
| 100% | 859,375 | **70.99** | **51.32** | **70.79** | **64.37** |

Table 16: The FT performance of using multiple layers of the teacher model for distillation. The attention maps of different layers are concatenated along the channel dimension. The teacher and student models are MAE-L and UNIP-S.

| Layer | SODA | MFNet-T | SCUT-Seg | Avg FT |
|---|---|---|---|---|
| 18 | **70.99** | **51.32** | **70.79** | **64.37** |
| 16+18 | 69.73 | 50.88 | 70.68 | 63.76 |
| 17+18 | 69.59 | 51.33 | 69.47 | 63.46 |
| 17+18+19 | 69.13 | 49.96 | 67.73 | 62.27 |

where Attention, FFN, LN refer to the self-attention module, the feedforward module, and the LayerNorm layer, respectively. Obviously, the self-attention module plays a crucial role in transforming the feature representation. The *global* attention pattern will bring the features of different tokens closer since different query tokens interact similarly with all key tokens. In contrast, the *local* attention pattern will make the features of different tokens further apart.

To investigate the relationships between features of different layers and models, we use the centered kernel alignment (CKA), a metric that measures the similarity between two feature maps. The details of CKA can refer to Kornblith et al. (2019). As shown in Fig. 9, features in the later layers of MAE-S, *e.g.*, the 10th and 11th layers, are similar to features in the shallow layers of DINO-S, *e.g.*, the 4th, 5th, and 6th layer, implying that the features of MAE-S are relatively lower level compared to DINO-S. **This is consistent with observations in Sec. 3.1 that the *local* attention patterns are distributed in the shallow layers of DINO-S, but are present in all layers of MAE.**

For MAE-L, the features in the middle layers (13th to 20th) exhibit high similarity with the middle-layer features of DINO-S (9th and 10th), due to the *hybrid* patterns in these layers. On the contrary, the features in the later layers (the 22nd and 23rd layers) gradually resemble the shallow layers of DINO-S, which can be attributed to the *local* patterns in the later layers of MAE-L.

It is noteworthy that the UNIP model effectively imitates the features in the middle layers of MAE-L. Its features align more closely with DINO-S than those of MAE-S, especially in the shallow and middle layers, demonstrating that attention distillation can implicitly change the features of distilled models like what feature distillation explicitly does.

## E   MORE EXPERIMENTS.

**Pre-training is important.** We compare the average FT performance of pre-trained and randomly initialized models. For pre-trained models, the performance is averaged across six different methods in Tab. 11. As shown in Tab. 17, models without pre-training consistently fall behind by 20.89% to 25.03%, regardless of model size. This gap widens with larger models, highlighting the importance of pre-training and the necessity of studying different pre-training approaches on infrared tasks.

Table 17: Comparison of initialization.

| Initialization | Tiny | Small | Base | Large |
|---|---|---|---|---|
| Random | 30.64 | 36.82 | 39.14 | 39.31 |
| Pre-training | **51.53** | **59.65** | **62.53** | **64.47** |
| | +20.89 | +22.83 | +23.39 | +25.16 |

Table 18: The fine-tuning performance of using different training strategies on the pre-training dataset InfMix.

| Model | Training Strategy | Epoch | SODA | MFNet-T | SCUT-Seg | Avg FT |
|---|---|---|---|---|---|---|
| MAE-Small | - | - | 63.36 | 42.44 | 60.38 | 55.39 |
| UNIP-Small | Separate Training (Stage1: RGB) | 100 | 68.98 | 50.01 | 68.79 | 62.59 |
| UNIP-Small | Separate Training (Stage2: Infrared) | 100 | 68.49 | **52.10** | 69.62 | 63.40 |
| UNIP-Small | Joint Training | 100 | **70.99** | 51.32 | **70.79** | **64.37** |

Table 19: The fine-tuning performance of using head-wise and layer-wise distillation.

| | Method | Layer (MAE-L) | Target | Avg NMI | SODA | MFNet-T | SCUT-Seg | Avg FT |
|---|---|---|---|---|---|---|---|---|
| 1 | Layer-wise | 18 | All 16 Heads | 0.1185 | **70.99** | 51.32 | 70.79 | 64.37 |
| 2 | | 18 | 5 (h), 6 (h), 8 (h), 10 (h), 13 (h), 15 (h) | 0.0985 | 70.37 | **52.01** | **71.82** | **64.73** |
| 3 | Head-wise | 18 | 2 (g), 3 (g), 4 (g), 7 (l), 9 (l), 14 (g) | 0.1049 | 68.75 | 50.89 | 70.43 | 63.36 |
| 4 | | 18 | 3 (g), 4 (g), 8 (h), 9 (l), 10 (h), 15 (h) | 0.1077 | 70.07 | 51.65 | 69.76 | 63.83 |
| 5 | Layer-wise | 24 | All 16 Heads | 0.1882 | 67.74 | 50.39 | 69.00 | 62.38 |
| 6 | Head-wise | 24 | 3 (h), 4 (h), 6 (h), 12(h), 15 (h), 16(h) | 0.1092 | **69.95** | **51.82** | **69.88** | **63.88** |

**Impact of the Size of the Pre-training Dataset.** Tab. 15 illustrates the fine-tuning performance with varying ratios of the InfMix dataset. A clear data scaling law is observed, where the performance consistently improves as the pre-training dataset size increases. This demonstrates the necessity of constructing the InfMix dataset, a much larger dataset than other infrared pre-training datasets like MSIP (Zhang et al., 2023) and Inf30 (Liu et al., 2024a). As we continue to expand the InfMix dataset, we can anticipate even greater advancements in model performance, potentially enabling breakthroughs in applications that rely on infrared data, such as autonomous driving (Xiong et al., 2021), and surveillance (Bondi et al., 2020).

**Impact of the Training Strategy on the Pre-training Dataset.** In Tab. 18, we conduct experiments involving a two-stage training process using MAE-L as the teacher model. In the first stage, the model is distilled using the RGB component of InfMix. In the second stage, the model is subsequently distilled using the infrared component of InfMix. As indicated in Tab. 18, benefiting from the hybrid pattern distillation, the model of the RGB training stage surpasses MAE-S by a large margin. After the infrared training stage, the model's average segmentation performance improves further. However, we observe a slight decline in performance on the SODA dataset. We attribute this to the problem of data distribution mismatch. Notably, half of the images in SODA depict indoor scenes, which are scarce in our infrared pre-training dataset InfPre. In contrast, such scenes are more prevalent in the ImageNet and COCO datasets. We believe this discrepancy also accounts for the inferior performance of the two-stage training compared to joint training. Joint training benefits from a wider data distribution, which contributes to improved generalization performance.

**Multi-layer Distillation.** In Tab. 16, we examine the use of attention maps from multiple layers of the teacher model for distillation. Interestingly, performance declines as more layers are included. We hypothesize that requiring a single student layer to mimic multiple teacher layers' attention maps introduces excessive complexity and noise, which impedes the distillation process. An adaptive selection of attention maps to minimize noise and redundancy could be a promising direction.

**Head-wise Distillation.** In Tab. 19, we explore the head-wise distillation, a more fine-grained distillation method. Compared to the layer-wise distillation in NMI-HAD, it directly utilizes different heads for distillation. The experimental setup involves using MAE-L (16 attention heads for each layer) as the teacher to distill UNIP-S (6 attention heads for each layer). First, we calculate the NMI for attention maps of each attention head in MAE-L and observe that not all attention heads within the same layer exhibit the same attention pattern. Therefore, we categorize these attention heads into three patterns: local (l), hybrid (h), and global (g). We then select six attention heads (the total number of heads in UNIP-S) as distillation targets. For the 18th layer of MAE-L, there are 5 global heads, 4 local heads, and 7 hybrid heads. We experiment with three different combinations: one containing only hybrid patterns (row 2), one containing only local and global patterns (row 3), and one containing all three patterns (row 4). The average NMI values for these combinations are comparable. Notably, the combination containing only hybrid attention patterns achieves the best performance, demonstrating the effectiveness of hybrid attention patterns even in head-wise distillation. Furthermore, using just 6 hybrid attention heads for distillation even surpasses the per-

Table 20: Details of the InfPre dataset. #Image and #Extraced image represent the number of original and extracted images from the dataset. Interval and Similarity denote the fixed-interval and similarity-based sampling methods, respectively. The value after the slash indicates the fixed interval or similarity threshold.

| Dataset | Task | Scenario | #Image | #Extracted Image | Average Width | Average Height | Sampling |
|---|---|---|---|---|---|---|---|
| RGBT-CC (Liu et al., 2021a) | Crowd Counting | Urban | 2,030 | 2,030 | 636 | 484 | - |
| KAIST (Hwang et al., 2015) | Object Detection | Driving | 95,328 | 9,546 | 640 | 512 | Interval / 10 |
| Infrared City (Yu et al., 2022) | Video Translation | Driving | 200,000+ | 20,187 | 256 | 256 | Interval / 10 |
| CVC-09 (cvc, 2016b) | Object Detection | Driving | 13,184 | 13,184 | 640 | 480 | - |
| CVC-14 (cvc, 2016a) | Object Detection | Driving | 8518 | 8518 | 640 | 471 | - |
| VAP (Palmero et al., 2016) | Semantic Segmentation | Indoors | 23,080 | 2,309 | 640 | 480 | Interval / 10 |
| RGBT-234 (Li et al., 2019) | Object Tracking | Surveillance | 117,612 | 11,762 | 628 | 459 | Interval / 10 |
| LTD (Nikolov et al., 2021) | Concept Drift | Surveillance | 26,820,000 | 15,749 | 384 | 288 | Similarity / 0.95 |
| Rain (Bahnsen & Moeslund, 2019) | Semantic Segmentation | Surveillance | 130,800 | 25,920 | 640 | 480 | Interval / 5 |
| Infrared Security (Liu et al., 2021b) | Object Detection | Surveillance | 8,999 | 8,999 | 495 | 386 | - |
| LLVIP (Jia et al., 2021) | Object Detection | Surveillance | 15,488 | 15,488 | 1,280 | 1,024 | - |
| LSOTB-TIR (Liu et al., 2020) | Object Tracking | Diverse | 600,000+ | 61,154 | 925 | 623 | Interval / 10 |
| Dual-Sensor (Chen et al., 2021a) | - | Driving | 73,638 | 14,728 | 384 | 288 | Interval / 5 |
| LasHeR (Li et al., 2022a) | Object Tracking | Diverse | 740,000+ | 74,035 | 879 | 554 | Interval / 10 |
| VT5000 (Tu et al., 2023) | Salient Object Detection | Diverse | 5,000 | 5,000 | 640 | 480 | - |
| Infrared Vehicle (Li et al., 2021b) | Object Detection | Driving | 13166 | 13,166 | 815 | 613 | - |
| Infrared Ship (Li & Wang, 2021) | Object Detection | Marine | 9,402 | 9,402 | 772 | 591 | - |
| DroneVehicle (Sun et al., 2022) | Object Detection | Aerial | 28,439 | 28,439 | 640 | 512 | - |
| Infrared Aerial (Liu et al., 2021c) | Object Detection | Aerial | 11,045 | 11,045 | 627 | 502 | - |
| VTUAV (Zhang et al., 2022) | Object Tracking | Aerial | 1,700,000 | 166,986 | 1,920 | 1,080 | Interval / 10 |
| M3FD (Liu et al., 2022) | Object Detection | Driving | 4,200 | 4,200 | 1,00,1 | 744 | - |
| OTCBVS IRIS Face (Abidi) | - | Human Face | 4,199 | 4,199 | 320 | 240 | - |
| Multispectral (Takumi et al., 2017) | Object Detection | Driving | 15,042 | 15,042 | 480 | 368 | - |
| InfPre | Pre-training | Diverse | - | 541,088 | 1,075 | 686 | - |

Table 21: The cosine similarity between pre-training and infrared segmentation datasets. The embeddings of images are extracted by DINO-B. The similarity is averaged over all pairwise images from different datasets.

| Pre-training dataset | Downstream dataset | | | |
|---|---|---|---|---|
| | SODA | MFNet-T | SCUT-Seg | Mean |
| ImageNet-1K (Deng et al., 2009) | 0.083 | 0.074 | 0.081 | 0.079 |
| COCO (Lin et al., 2014) | 0.111 | 0.101 | 0.106 | 0.106 |
| InfMix | 0.200 | 0.227 | 0.236 | 0.221 |
| InfMix (gray) | **0.216** | **0.246** | **0.254** | **0.239** |

formance of distilling all 16 heads in the 18th layer (row 1). This phenomenon is also observed in the 24th layer. This suggests that there may be redundancy in the attention maps within a single layer. Therefore, we believe that more fine-grained distillation, such as head-wise distillation, is a highly promising research direction.

# F PRE-TRAINING DATASET.

## F.1 THE INFPRE DATASET.

The InfPre dataset is constructed by collecting images from 23 infrared-related visual datasets. The details of the extracted datasets are presented in Tab. 20. To reduce the redundancy in images with similar backgrounds, we employ two sampling methods: fixed-interval sampling and similarity-based sampling. For datasets containing diverse image sequences with different backgrounds, frames are sampled at fixed intervals (*e.g.* 2, 5, and 10) within each sequence. For datasets captured in the same location, we only sample frames that are less similar to each other. The cosine similarity of image embeddings extracted by DINO-B is used as the similarity metric. Images with high similarity to those already sampled images will be discarded. The Faiss (Johnson et al., 2021) library is utilized to accelerate the sampling process.

### F.2 THE INFMIX DATASET.

The InfMix dataset combines the InfPre, the subset of ImageNet-1k (Deng et al., 2009), and the training set of COCO (Lin et al., 2014), totaling 859,375 images. Tab. 21 compares the similarity between various pre-training datasets and three infrared segmentation datasets used in our benchmark. Notably, compared to RGB datasets like ImageNet-1k and COCO, the mixed dataset exhibits higher similarity with infrared downstream tasks, thereby mitigating the representation shift between pre-training and downstream data. Moreover, converting RGB images to grayscale further enhances this similarity, resulting in better fine-tuning performance, as shown in Tab. 8.

## G    MORE VISUALIZATIONS.

We present visualizations of distilled models in this section. Fig. 10 shows the attention maps of different supervised and CL methods of various sizes. As shown in Fig. 11, the deep layers of UNIP-S exhibit *hybrid* patterns, indicating that UNIP effectively transfers these patterns from the teacher to the student. Additionally, compared to MAE-L, attention maps in UNIP-S focus more on shape information than textures, as evident in Fig. 12. The emergence of *hybrid* patterns in deep layers and the reduced bias towards texture both contribute to the excellent performance of UNIP. The attention maps of RGB image inputs are visualized in Fig. 13, exhibiting nearly identical attention pattern distribution with infrared images in Fig. 3.

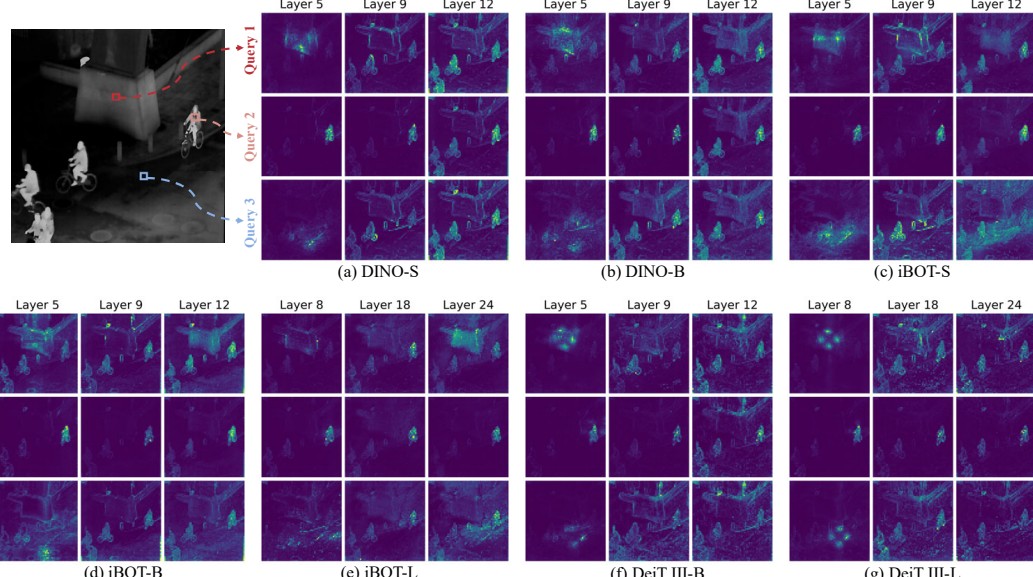

Figure 10: Visualizations of attention maps in supervised and CL models. The attention maps are averaged over different heads. All CL and supervised methods share similar attention pattern distribution across layers.

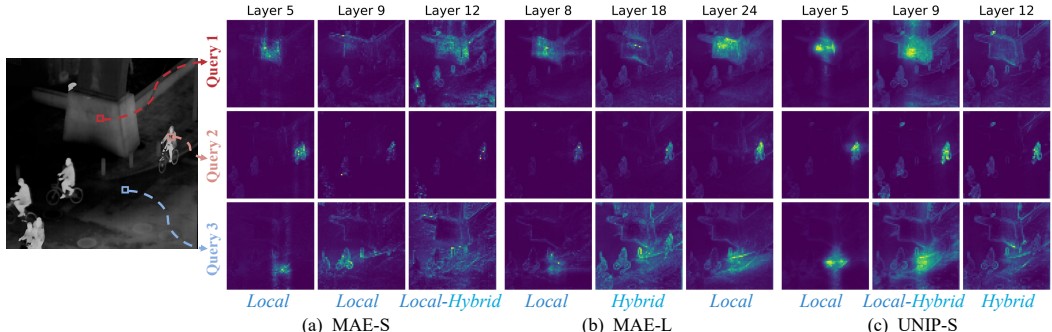

Figure 11: Visualizations of layerwise attention maps in MAE and UNIP-S distilled from MAE-L. The *hybrid* patterns emerge in the later layers of UNIP-S but in the middle layers of MAE-L.

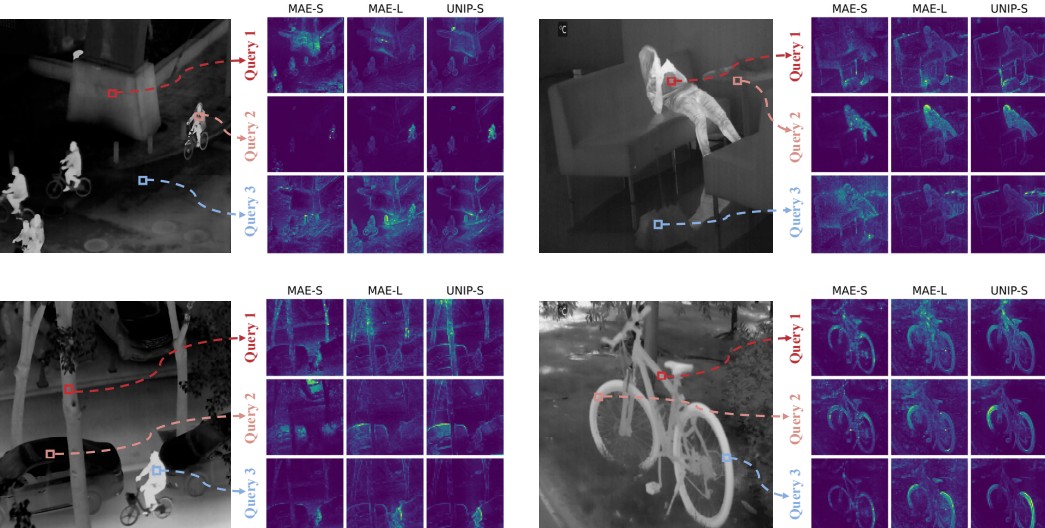

Figure 12: Visualizations of attention maps in MAE and UNIP distilled from MAE-L. Attention maps from the 12th layer of MAE-S, the 18th layer of MAE-L, and the 12th layer of UNIP-S are displayed, respectively. Compared to MAE-S and MAE-L, UNIP-S exhibits reduced texture bias, emphasizing shape information over textures.

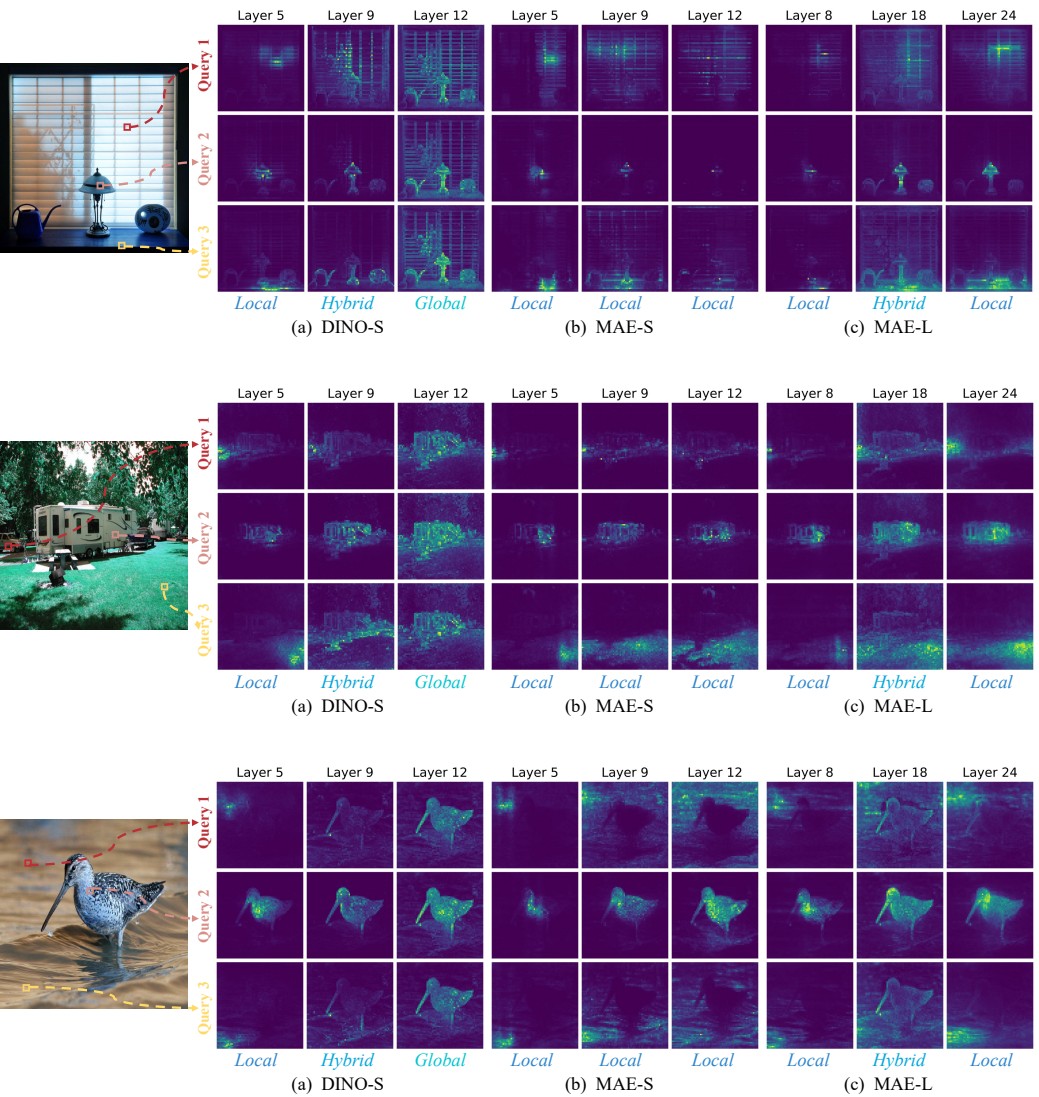

Figure 13: Attention maps of RGB image inputs for different query tokens in three representative layers. Each query token's attention map corresponds to a row in the attention matrix, averaged over different heads. Images are from ImageNet (Deng et al., 2009).

