# OpenReview forum: "UNIP: Rethinking Pre-trained Attention Patterns for Infrared Semantic Segmentation"
_ICLR.cc/2025/Conference — ICLR 2025 Poster_

### Official Review · Reviewer_gRqm · 2024-10-24

**Soundness:** 4
**Presentation:** 4
**Contribution:** 3
**Rating:** 8
**Confidence:** 5

**Summary:**

This work first deeply analyzes the transfer performance on infrared tasks of popular pre-training regimes, including supervised training, contrastive learning and masked image modeling. Based on three valuable observations, this work proposes a pre-training and finetuning framework for infrared image segmentation, dubbed UNIP—specifically, an NMI-HAD distillation algorithm, a mixed large-scale InfMix dataset, and an LL-FPN architecture. Complete experiments and analysis demonstrate the effectiveness of the proposed algorithm.

**Strengths:**

This work is well-motivated, clearly written and very convincing. All observations and hypotheses are proved by empirical or numerical evidence, which makes the whole work complete and sound.

**Weaknesses:**

Overall, I feel pretty OK with this work, and there are no **MAJOR** weaknesses. Some minor suggestions are as follows for the author's reference.

- As this work focuses only on infrared images, the word infrared should appear in the title to make it sound and avoid misunderstanding, *e.g.*, for infrared semantic segmentation.
- Some abbreviations should be given their corresponding specific names and appropriate citations if necessary when they first appear, *e.g.*, SSL (semi-supervised learning) in Sec. 2.1.
- Unalighed letters of ViT in Fig. 8.

**Questions:**

Please refer to the WEAKNESSES part.

**Details Of Ethics Concerns:**

An Ethics Review may be needed for the claimed InfMix data, *e.g.*, human subjects.

---

> ### Author Response · Authors · 2024-11-21
> **Response to Reviewer gRqm**
>
> Thanks for your effort in reviewing our paper and giving detailed suggestions. We are very pleased to receive such high praise for our work, including the motivation, writing, methods, and experiments. We sincerely appreciate it! We hope the following responses can solve your concerns.
>
> > Q1: About the title.
>
> Thanks for your valuable suggestion.  We modify the title from "Semantic Segmentation" to "Infrared Semantic Segmentation" in our revised draft. Additionally, the NMI-HAD and LL-FPN components of our method are also applicable to other image modalities, such as RGB. For further details on this aspect, please refer to our response to Reviewer FdMz, specifically in Q1.
>
> > Q2: About the abbreviations.
>
> Thanks for your detailed suggestion. We recognize that there is a lexical error. Given that our research encompasses both supervised and self-supervised pre-training methods, we replace "SSL" with "pre-training" in line 146 of the revised draft and include citations of both supervised and self-supervised approaches.
>
> > Q3: About the unaligned letters in Fig. 8.
>
> Thanks for your meticulous finding. We correct this typo in the revised draft.
>
> > Q4: About the Ethics Review.
>
> Our constructed dataset InfMix is derived from the extraction and integration of 25 publicly accessible RGB and infrared datasets. These datasets are available for public use. During the extraction process, we employ data deduplication methods only, ensuring that no personal biases were introduced. We adhere to all usage requirements of the original datasets and use them solely for scientific research purposes. Importantly, the dataset does not include any additional metadata or labels that could be used to identify individuals. Detailed information regarding the composition and construction process of our dataset is provided in Appendix E of our paper.  In accordance with the usage agreements of the original datasets, when publicly sharing our dataset, we will only release the index list corresponding to the images we utilized, without redistributing the original datasets.
>
> ---
>
> We would be more than happy to discuss any further questions!

---

### Official Review · Reviewer_nH3J · 2024-10-30

**Soundness:** 3
**Presentation:** 3
**Contribution:** 2
**Rating:** 5
**Confidence:** 4

**Summary:**

This paper presents a deep analysis on why RGB pre-trained models performs worse on thermal images. Based on the analysis, the authors present a NMI-guided distillation for improved segmentation. Experiments on different datasets are performed.

**Strengths:**

- This paper presents a deep analysis on benchmarking the infrared semantic segmentation performance of various pre-training models.
- The experimental results are extensive, and the writings are good.

**Weaknesses:**

- The title of the paper is inappropriate as it misleadingly suggests a general semantic segmentation task. In fact, this paper mainly focuses on infrared segmentation of small models by adapting pre-trained models.
- According to the claims of the authors, the MIM (Masked Image Modeling) strategy focuses more on texture information and is not suitable for segmenting infrared images. Why using an MAE-based model as the teacher for distillation can achieve better results?
- This paper seems no comparison with existing distillation strategies used in domain-adaptive methods. If possible, I think it is better to give a comparison.
- More experiments on built datasets can be given to demonstrate the effectiveness of proposed method. For example, we first train on RGB images and then fine-tune on infrared image, or jointly train on both dataset.

**Questions:**

Please answer the title, claim, and experiment questions in weakness

---

> ### Author Response · Authors · 2024-11-21
> **Response to Reviewer nH3J (1/3)**
>
> Thanks for your effort in reviewing our paper and providing insightful suggestions. We hope the following responses can address your concerns.
>
> > Q1: About the title.
>
> Thanks for your constructive suggestion. Our primary motivation is to enhance the performance of the infrared semantic segmentation task from the perspective of pre-trained models. Therefore, **in the revised draft, we update the title from "… Semantic Segmentation" to "… Infrared Semantic Segmentation".** Additionally, the NMI-HAD and LL-FPN components of our method are also applicable to other image modalities, such as RGB. For further details on this aspect, please refer to our response to Reviewer FdMz, specifically in Q1.
>
> > Q2: About the texture bias in distillation.
>
> This is a noteworthy question. In this paper, we elucidate two key factors that enhance the performance of pre-trained models in infrared semantic segmentation. The first factor is the presence of hybrid attention patterns, and the second factor is the reduction of biases towards texture. Therefore, the reasons for improved performance are two-fold. For example, using MAE-Large as the teacher to distill UNIP-Small.
>
> - Firstly, UNIP-Small exhibits hybrid attention patterns in the deep layers through the NMI-HAD method, as illustrated in Fig. 11 of our paper, leading to better segmentation performance.
> - Secondly, we utilize 541,088 infrared images for the distillation of UNIP-Small, enabling the model to adapt to the characteristics of weak textures in infrared images. As a result, the distilled UNIP-Small significantly reduces its bias toward texture information, as illustrated in Fig. 12 of our paper, leading to better transfer performance on infrared datasets.
>
> When comparing UNIP-Small with MAE-Small, both reasons are significant. However, in comparison with MAE-Large, we believe that the second reason may have a greater impact, as the hybrid pattern is also present in MAE-Large. Additionally, when compared to models distilled by other models, such as DINO-Base, we think that the quality of attention maps produced by the teacher model is a critical factor. Generally, larger models tend to generate higher-quality attention maps. Therefore, models distilled from MAE-Large and iBOT-Large demonstrate superior performance compared to those distilled from DINO-Base.

---

> ### Author Response · Authors · 2024-11-21
> **Response to Reviewer nH3J (2/3)**
>
> > Q3: Comparison with distillation strategies in domain-adaptive methods.
>
> Thank you for your suggestions. We think that our work differs from domain-adaptive methods in the following three main aspects:
>
> - **Different Application Scenarios**: Domain-adaptive methods are typically tailored for specific downstream tasks and corresponding network structures, such as semantic segmentation [1,2] and object detection [3,4,5], with the requirement that the target labels in the source and target domains should overlap as much as possible. In contrast, our study focuses on adapting pre-trained models without being tied to any specific visual task. The adapted model can be applied to various visual tasks.
>
> - **Different Implementation Approaches**: In domain-adaptive methods, the source model is usually trained on labeled data from the source domain, which equips it with the basic ability to perform specific tasks like semantic segmentation. The model is then adapted using labeled/unlabeled data from the target domain, allowing the adapted target model to perform these tasks without further fine-tuning. In contrast, our approach involves a source ViT model pre-trained on labeled/unlabeled datasets from the source domain, which is unable to complete specific tasks. We then adapt the pre-trained model using unlabeled data from the target domain. However, the adapted pre-trained model still lacks the ability to perform specific tasks and requires supervised fine-tuning with task-specific networks and labeled datasets.
>
> - **Different Evaluation Methods**: In domain-adaptive methods, the adapted model is evaluated directly on the test set. Conversely, in our approach, after adapting the pre-trained model, we conduct supervised fine-tuning on downstream tasks for evaluation.
>
> Regarding other distillation methods, in our early experiments, we tried several different approaches, including masking the input images followed by feature or attention distillation, as well as distilling multiple layers from the teacher to multiple layers of the student. However, these methods did not perform as well as our current approach. To be honest, we have not identified suitable distillation methods within the domain-adaptive framework to compare against. If you have any suggested methods, we would be very interested in conducting experiments in this area.
>
> [1] *Hoyer L, Dai D, Wang H, et al. MIC: Masked image consistency for context-enhanced domain adaptation. CVPR, 2023.*
>
> [2] *Wang K, Kim D, Feris R, et al. CDAC: Cross-domain attention consistency in transformer for domain adaptive semantic segmentation. ICCV, 2023.*
>
> [3] *Cao S, Joshi D, Gui L Y, et al. Contrastive mean teacher for domain adaptive object detectors. CVPR, 2023.*
>
> [4] *Li J, Xu R, Ma J, et al. Domain adaptive object detection for autonomous driving under foggy weather. WACV, 2023.*
>
> [5] *Weng W, Yuan C. Mean Teacher DETR with Masked Feature Alignment: A Robust Domain Adaptive Detection Transformer Framework. AAAI, 2024.*

---

> ### Author Response · Authors · 2024-11-21
> **Response to Reviewer nH3J (3/3)**
>
> > Q4: Comparison between independent training and joint training.
>
> Thanks for your valuable question. We conduct experiments involving a two-stage training process using MAE-Large as the teacher model. In the first stage, the model is distilled using the RGB component of InfMix, i.e., the subset of ImageNet and the training set of COCO. In the second stage, the model is subsequently distilled using the infrared component of InfMix, i.e., the constructed InfPre dataset.
>
> As indicated in the table below, benefiting from the hybrid pattern distillation, the model of the RGB training stage surpasses MAE-Small by a large margin. After the infrared training stage, the model's average segmentation performance improves further, attributed to both the hybrid pattern distillation and the mitigation of distribution shift issues for infrared images. However, we observe a slight decline in performance on the SODA dataset. We attribute this to the problem of data distribution mismatch. Notably, half of the images in SODA depict indoor scenes, which are scarce in our infrared pre-training dataset InfPre. In contrast, such scenes are more prevalent in the ImageNet and COCO datasets. We believe this discrepancy also accounts for the inferior performance of the two-stage training compared to joint training. Joint training benefits from a wider data distribution, which contributes to improved generalization performance.
>
> |   Model    |        $\quad\quad\quad\quad\quad\enspace$       Training               | Epoch |   SODA    |   MFNet   | SCUT-Seg  |  Avg FT   |
> | :--------: | :----------------------------------: | :---: | :-------: | :-------: | :-------: | :-------: |
> | MAE-Small  |                  -                   |   -   |   63.36   |   42.44   |   60.38   |   55.39   |
> | UNIP-Small |   Separate Training (Stage1: RGB)    |  100  |   68.98   |   50.01   |   68.79   |   62.59   |
> | UNIP-Small | Separate Training (Stage2: Infrared) |  100  |   68.49   | **52.10** |   69.62   |   63.40   |
> | UNIP-Small |        Joint Training (ours)         |  100  | **70.99** |   51.32   | **70.79** | **64.37** |
>
> ---
>
> We would be more than happy to discuss any further questions!

---

> ### Comment · Area_Chair_PEMD · 2024-11-25
>
> Dear Reviewer nH3J,
>
> Could you kindly review the rebuttal thoroughly and let us know whether the authors have adequately addressed the issues raised or if you have any further questions.
>
> Best,
>
> AC of Submission2787

---

> > ### Comment · Reviewer_nH3J · 2024-11-27
> >
> > About Q2, I still believe there are some conflicts between the claim that "the MIM (Masked Image Modeling) strategy focuses more on texture information and is not suitable for segmenting infrared images" and the method using MAE-L as the teacher.

---

> > > ### Author Response · Authors · 2024-11-28
> > >
> > > We appreciate your concern regarding the texture bias. In our paper, we employ multiple pre-trained models as teachers, including MAE-Large, DINO-Base, and iBOT-Large.
> > >
> > > **Firstly, the student model distilled from MAE-Large does not consistently outperform the other models.** The performance comparison of using the equally sized MAE-Large and iBOT-Large as teacher models is presented in the table below (sourced from Tab. 4 and Tab. 11 of the paper). For student models of various sizes, those distilled from MAE-Large demonstrate comparable performance to those from iBOT-Large in the fine-tuning metric. **However, they consistently lag behind those from iBOT-Large in the linear probing metric.** As discussed in lines 903-909 of the paper, the linear probing metric provides a more direct reflection of the pre-trained model’s generalization ability for downstream tasks, as it does not alter the parameters of the pre-trained model. Therefore, the performance gap observed in linear probing indicates that models distilled from MAE-Large exhibit less generalization ability in infrared tasks compared to those from iBOT-Large. We believe that the texture bias inherent in MAE models is a key factor contributing to this phenomenon, which aligns with our assertion in Sec. 4 that “the texture bias hinders the model’s generalization ability on infrared images.” In the fine-tuning metric, the adjustment of pre-trained parameters with labeled infrared datasets reduces this generalization gap, leading to comparable fine-tuning performance.
> > >
> > > | Student Model | Teacher Model | Average Fine-tuning | Average Linear Probing |
> > > | :-----------: | :-----------: | :-----------------: | :--------------------: |
> > > |   UNIP-Tiny   |   MAE-Large   |   60.27  (-0.30)    |     35.82  (-5.24)     |
> > > |   UNIP-Tiny   |  iBOT-Large   |      **60.57**      |       **41.06**        |
> > > |  UNIP-Small   |   MAE-Large   |   64.37  (-0.33)    |     44.04  (-4.33)     |
> > > |  UNIP-Small   |  iBOT-Large   |      **64.70**      |       **48.37**        |
> > > |   UNIP-Base   |   MAE-Large   | **65.28**  (+0.21)  |     47.43  (-4.15)     |
> > > |   UNIP-Base   |  iBOT-Large   |        65.07        |       **51.58**        |
> > >
> > > **Secondly, the inclusion of infrared images in the pre-training dataset also narrows the gap between MAE-Large and other teacher models.** To further investigate the reasons behind the comparable fine-tuning performance of MAE-Large and iBOT-Large, we conduct additional ablation experiments. We utilize ImageNet-1K as the pre-training dataset and employ both iBOT-Large and MAE-Large to distill UNIP-Small. The performance of student models is presented in the table below. Compared to the results obtained with InfMix, using ImageNet-1K as the pre-training dataset significantly increases the performance gap between models distilled from MAE-Large and those distilled from iBOT-Large, both in fine-tuning (-1.14 vs -0.33) and linear probing (-5.97 vs -4.33).  This comparison indicates that the infrared data in InfMix suppress the transfer of texture bias from the MAE teacher model to the student model during the distillation process, thereby narrowing the gap between MAE-Large and other models when used as teacher models. Therefore, under the combined influence of infrared pre-training data and fine-tuning with infrared labeled data, models distilled from MAE-Large achieve fine-tuning performance comparable to those from iBOT-Large.
> > >
> > > | Pre-training Dataset | Teacher Model | Student Model | SODA  | MFNet | SCUT-Seg |   Average FT   | SODA  | MFNet | SCUT-Seg |   Average LP   |
> > > | :------------------: | :-----------: | :-----------: | ----- | ----- | -------- | :------------: | ----- | ----- | -------- | :------------: |
> > > |        InfMix        |   MAE-Large   |  UNIP-Small   | 70.99 | 51.32 | 70.79    | 64.37  (-0.33) | 55.25 | 33.49 | 43.37    | 44.04  (-4.33) |
> > > |        InfMix        |  iBOT-Large   |  UNIP-Small   | 70.75 | 51.81 | 71.55    |   **64.70**    | 60.28 | 37.16 | 47.68    |   **48.37**    |
> > > |     ImageNet-1K      |   MAE-Large   |  UNIP-Small   | 69.39 | 49.11 | 69.63    | 62.71  (-1.14) | 51.96 | 30.23 | 40.05    | 40.75  (-5.97) |
> > > |     ImageNet-1K      |  iBOT-Large   |  UNIP-Small   | 70.45 | 50.53 | 70.57    |   **63.85**    | 59.31 | 35.57 | 45.27    |   **46.72**    |
> > >
> > > In our paper, to illustrate the generalizability of the proposed method, we conduct experiments using multiple teacher models. We believe that using iBOT-Large as a teacher model may be a better choice than MAE-Large, as models distilled from iBOT-Large demonstrate strong performance in both fine-tuning and linear probing.
> > >
> > > Thank you again for your valuable feedback. We are open to your further response and instructions. Additionally, we have uploaded a newly revised draft and added the two-stage training in Appendix E (Tab. 18 and Lines 1159-1169).

---

> > > > ### Author Response · Authors · 2024-12-02
> > > >
> > > > Dear Reviewer nH3J,
> > > >
> > > > We sincerely appreciate your valuable feedback and hope that our responses have effectively addressed your concerns. As the discussion period deadline approaches, we would be grateful if you could let us know if you have any additional questions. We are eager to address any remaining issues.
> > > >
> > > > Best regards,
> > > >
> > > > The Authors

---

### Official Review · Reviewer_FdMz · 2024-11-02

**Soundness:** 3
**Presentation:** 2
**Contribution:** 3
**Rating:** 5
**Confidence:** 3

**Summary:**

This paper leverages strong pre-trained models to boost the performance of infrared semantic segmentation. The authors identify the best attention patterns and propose the hybrid-attention distillation to improve the performance and efficiency at the same time. A large-scale mixed dataset is also presented for pre-training. The proposed methods achieve the new state-of-the-art results.

**Strengths:**

1. The benchmark of vision foundation models on infrared segmentation datasets and the introduction of NMI to identity attention patterns for distillation contribute to the community.
2. Extensive and comprehensive experiments are conducted and the proposed approach significantly surpasses state-of-the-art infrared or RGB segmentation methods.

**Weaknesses:**

1. The underlying motivation is not very clear. In my opinion, the authors sometimes aim to emphasize the performance in infrared semantic segmentation and sometimes aim to emphasize the versatility of the proposed method. However, on the one hand, the proposed approach is not specific to infrared semantic segmentation tasks. On the other hand, from the viewpoint of visual pre-training and downstream transfer learning, there is a lack of experiments on more general datasets and performance comparisons with related works. Compared to previous works in knowledge distillation, the biggest contribution may be to provide a method to choose which layer to distill. This is the main concern for me.

2. It is strange to distinguish between the last two attention patterns by calling them “local-global” and “global”. The “global” attention patterns are just attention collapse as the authors stated. Therefore, if only considering a single distillation target, the last layer may not be the best choice. I believe previous works have suggested this. If using multiple hierarchical features which is more common, the conclusion may be different.

**Questions:**

For NMI-HAD, is only the attention map of a single layer used for distillation?

---

> ### Author Response · Authors · 2024-11-21
> **Response to Reviewer FdMz (1/2)**
>
> Thanks for your effort in reviewing our paper and giving kind suggestions. We hope the following responses can solve your concerns.
>
> > Q1: About the underlying motivation and contribution.
>
> Motivation:
>
> Our motivation is to explore the transfer performance on infrared segmentation tasks of various RGB pre-trained models, and further improve their infrared segmentation performance based on some insightful findings. Our work’s Chain-of-Thought (CoT) is illustrated in Fig. 1 of the paper, following the sequence from step1 to step3. We first evaluate the infrared segmentation performance of different pre-trained models and then find two key phenomena: (1) The hybrid attention pattern is essential for semantic segmentation.  (2) The texture bias hinders the model’s generalization on infrared images. For the first phenomenon, we introduce the NMI-HAD and LL-FPN to utilize the importance of the hybrid attention pattern. For the second phenomenon, we construct a large-scale dataset InfMix for pre-training to mitigate the distribution shift issue. **Notably, all the proposed methods and datasets are dedicated to improving the infrared segmentation performance, which is our primary goal.** In the revised draft, we have changed the title of our paper from "… Semantic Segmentation" to "… Infrared Semantic Segmentation" to better emphasize our motivation and to prevent any potential misunderstandings.
>
> **The applicability of the proposed methods to other infrared visual tasks and image modalities is a bonus.**  (1) The UNIP can be applied to other infrared visual tasks, such as object detection, since UNIP focuses on enhancing segmentation performance from the perspective of pre-training, without relying on architecture specifically designed for segmentation tasks.  (2) The NMI-HAD and LL-FPN are applicable to other image modalities like RGB and depth, since the hybrid attention pattern is also essential in these modalities, as indicated in Tab. 10 of our paper. Moreover, we distill the hybrid pattern from MAE-Large on the ImageNet-1K dataset using NMI-HAD and fine-tune the distilled model on ADE20K using LL-FPN. The results in the table below demonstrate the effectiveness of NMI-HAD and LL-FPN in RGB datasets.
>
> |   $\quad$Model    | Layer | ADE20K mIoU |
> | :--------: | :---: | :---------: |
> | MAE-Small  |   -   |    41.5     |
> | iBOT-Small |   -   |    45.4     |
> | UNIP-Small |  18   |  **48.2**   |
> | UNIP-Small |  24   |    46.9     |
>
> Contribution:
>
> There are three main differences between our paper and previous works [1,2] in knowledge distillation for pre-trained models.
>
> 1. We evaluate the transfer performance of six popular pre-training methods, including supervised training, contrastive learning, and masked image modeling, while previous works mainly focus on a single pre-training method.
>
> 2. We conduct a thorough analysis of the performance differences among various pre-trained models, revealing distinct distributions of attention patterns across these models and correlating them with their pre-training tasks. These valuable findings provide insight into the intrinsic characteristics of different pre-training methods and models, as appreciated by Reviewers ctU1, nH3J, and gRqm.
>
> 3. We perform the distillation in the domain transfer setting, while previous works mainly focus on the RGB domain. Our work demonstrates that selective knowledge distillation, combined with unlabeled domain-specific data, can significantly unleash the potential of the large models pre-trained on extensive RGB datasets. It offers a viable route to improve the performance on tasks in domains with insufficient labeled data, from the perspective of model pre-training.
>
> We believe that our analysis and experiments can catalyze further exploration, encouraging more researchers to focus on and continue improving this area. For instance, future work could involve multi-layer distillation, multi-model distillation, or more fine-grained head-wise distillation, as suggested by Reviewer ctU1.
>
> [1] *Bai Y, Wang Z, Xiao J, et al. Masked autoencoders enable efficient knowledge distillers. CVPR, 2023.*
>
> [2] *Xiong Y, Varadarajan B, Wu L, et al. Efficientsam: Leveraged masked image pretraining for efficient segment anything. CVPR, 2024.*

---

> ### Author Response · Authors · 2024-11-21
> **Response to Reviewer FdMz (2/2)**
>
> > Q2: About the attention patterns.
>
> Thanks for your valuable questions. We respond to them one by one.
>
> - The attention patterns in this paper are classified into local, hybrid, and global. These three attention patterns are qualitatively summarized by the visualization of attention maps (Sec. 3.1) and are quantitatively measured by NMI (Sec. 3.2).  Our paper does not describe the attention pattern as “local-global”, do you mean to refer to “hybrid”? As shown in Fig. 3 (a), in the hybrid attention pattern, query tokens attend to both nearby and foreground tokens, while in the global pattern, all query tokens focus on the same foreground tokens, resulting in nearly identical attention maps.
>
> - Previous works like DMAE [1] indeed suggest that the last layer may not be the best choice for single-layer distillation. However, their findings are limited to a specific type of pre-trained model and do not provide an in-depth analysis of why the last layer is suboptimal. On the contrary, our study thoroughly analyzes the representation of different pre-trained models, revealing that the underlying reasons for the last layer's inefficacy may differ significantly across models. For example,  MAE models exhibit local patterns in the final layer, while DINO models exhibit global patterns. Moreover, previous works are unable to identify the layer best suitable for distillation, while we provide a straightforward yet effective method to address this issue.
>
> - For multiple hierarchical feature distillation, we conduct experiments as shown in Tab.16 and discuss them in lines 1109-1113 of our paper. We also perform additional experiments, and the consolidated results are presented in the table below. The experimental results indicate that increasing the number of layers used for distillation leads to a decline in performance. As stated in our paper, we believe that employing multiple layers for distillation complicates the distillation objectives and introduces unnecessary redundancy, thereby negatively impacting the distillation performance. An adaptive selection of attention maps might be a potential solution to reduce noise and redundancy. Combined with the head-wise distillation suggested by Reviewer ctU1, the selection and aggregation of head-wise attention maps in multiple layers is precisely one of the directions we intend to explore in the future.
>
>   |  $\enspace$Layers   |   SODA    |  MFNet-T  | SCUT-Seg  |  Avg FT   |
>   | :-------: | :-------: | :-------: | :-------: | :-------: |
>   | 18 (ours) | **70.99** |   51.32   | **70.79** | **64.37** |
>   |   16+18   |   69.73   |   50.88   |   70.68   |   63.76   |
>   |   17+18   |   69.59   | **51.33** |   69.47   |   63.46   |
>   |   18+24   |   69.51   |   49.93   |   69.37   |   62.94   |
>   | 17+18+19  |   69.13   |   49.96   |   67.73   |   62.27   |
>
> > Q3: Whether only the attention map of a single layer is used for distillation?
>
> Yes, only the attention map of a single layer is used for distillation. As discussed in response to Q2, vanilla multi-layer distillation does not perform as well as single-layer distillation.
>
> ---
>
> We would be more than happy to discuss any further questions!
>
> [1] *Bai Y, Wang Z, Xiao J, et al. Masked autoencoders enable efficient knowledge distillers. CVPR, 2023.*

---

> > ### Comment · Reviewer_FdMz · 2024-11-28
> > **Response to rebuttal**
> >
> > Thanks for your comments. However, I still feel that the methods proposed in this work or the conclusions drawn from the analysis are not unique to infrared images. In other words, I can't find a reasonable reason why the proposed methods should be used on infrared images instead of general natural images.

---

> > > ### Author Response · Authors · 2024-11-29
> > >
> > > Thank you for your valuable feedback. We understand your concerns. From your perspective, you believe that the method proposed in the paper could be applied to natural images, while we have focused on its application to infrared semantic segmentation tasks.
> > >
> > > However, considering our research trajectory, **our core motivation is to enhance the performance of infrared semantic segmentation tasks.** Previous approaches [1,2] have aimed to achieve this by designing network architectures specifically tailored for infrared semantic segmentation, whereas our work explores an alternative approach: optimizing pre-trained models specifically for infrared semantic segmentation tasks to improve performance. To facilitate this exploration, our work is divided into three stages: benchmark establishment, cause analysis, and method proposal, as illustrated in Fig. 1 of the paper.
> > >
> > > (1) In the benchmark establishment stage (Sec. 2), we evaluate the performance of six RGB pre-training methods on **three infrared semantic segmentation datasets**. Our findings reveal that the performance of pre-trained models on ImageNet did not correlate with their performance on **infrared segmentation tasks** (Tab. 1). Furthermore, we observe that supervised and contrastive learning methods demonstrate superior generalization abilities for **infrared segmentation tasks** compared to masked image modeling approaches (Fig. 2).  (2) In the causal analysis stage (Sec. 3), to analyze the performance discrepancies among various pre-training methods on **infrared segmentation tasks**, we conduct an in-depth analysis of the attention maps from the pre-trained models. Our findings indicate that the existence of hybrid attention patterns (Fig. 3 - Fig. 5) and the reduced bias towards texture (Tab. 2) both play crucial roles in enhancing the performance of pre-trained models in **infrared segmentation tasks**. (3) In the method proposal stage (Sec. 4), based on the observations and analyses from the previous two sections, we propose UNIP, a framework designed to improve the **infrared segmentation performance** of pre-trained models through three key aspects: the pre-training objective (NMI-HAD), the pre-training data (InfMix), and the fine-tuning architecture (LL-FPN). We also discussed this matter with Reviewer ctU1 and have elaborated on the relationship between the paper's motivation and the proposed methods in Appendix B of the revised draft.
> > >
> > > **It is evident that the three stages of our work align with our research motivation, focusing on the model’s performance on infrared semantic segmentation tasks. Our conclusions are grounded in experimental explorations conducted on infrared segmentation datasets, which naturally leads us to apply the proposed methods to these tasks, fully in line with our motivation.**
> > >
> > > Certainly, we acknowledge that certain aspects of the proposed methods and conclusions, such as NMI-HAD and LL-FPN, can be applied to other domains, including RGB and depth maps. We have highlighted this in Tab. 10 (Lines 505-515) and the future work section (Lines 535-539) of the paper, presenting it as a potential avenue for future research and extensions of our work.
> > >
> > > We believe that the scope of content that a paper can encompass is limited, and it is not feasible for us to conduct experiments across all domains. **Our initial motivation was to enhance performance in infrared segmentation tasks. Therefore, we identify key issues based on our experiments on infrared segmentation datasets and propose corresponding improvement methods specifically for these tasks.** Additionally, we demonstrate the applicability of these methods to other image domains. From a scientific research perspective, we consider this to be a reasonable research trajectory. **Overall, we believe that the applicability of our proposed methods or the conclusions to other image domains should be viewed as an advantage rather than a deficiency.**
> > >
> > > Thank you again for your feedback. We are open to your further response and instructions.
> > >
> > > ---
> > >
> > > [1] *Chen J, Bai X. Atmospheric transmission and thermal inertia induced blind road segmentation with a large-scale dataset tbrsd. ICCV, 2023.*
> > >
> > > [2] *Li C, Xia W, Yan Y, et al. Segmenting objects in day and night: Edge-conditioned CNN for thermal image semantic segmentation. TNNLS, 2021.*

---

> > > > ### Author Response · Authors · 2024-12-02
> > > >
> > > > Dear Reviewer FdMz,
> > > >
> > > > We sincerely appreciate your valuable feedback and hope that our responses have effectively addressed your concerns. As the discussion period deadline approaches, we would be grateful if you could let us know if you have any additional questions. We are eager to address any remaining issues.
> > > >
> > > > Best regards,
> > > >
> > > > The Authors

---

> ### Comment · Area_Chair_PEMD · 2024-11-25
>
> Dear Reviewer FdMz,
>
> Could you kindly review the rebuttal thoroughly and let us know whether the authors have adequately addressed the issues raised or if you have any further questions.
>
> Best,
>
> AC of Submission2787

---

### Official Review · Reviewer_ctU1 · 2024-11-04

**Soundness:** 3
**Presentation:** 4
**Contribution:** 3
**Rating:** 6
**Confidence:** 4

**Summary:**

This paper focuses on the pre-trained attention patterns for semantic segmentation in the infrared domain. It first concludes from the performance of different pre-training methods in infrared segmentation that the pretraining task is the key factor. Then, it takes deeper research on the attention pattern and states the importance of the hybrid attention pattern. Based on this insight, this paper used NMI to indicate the attention pattern and proposed a new pre-training framework named NMI-HAD to distill the layers with hybrid attention patterns. It also proposes LL-FPN and has improved fine-tuning performance

**Strengths:**

1.	The writing is easy to follow.
2.	The ideas proposed by the author are well supported by the detailed experimental results.
3.	The pre-training method is important in semantic segmentation.

**Weaknesses:**

1. Although the analysis of attention patterns is persuasive, more comparison with the attention patterns in other RGB semantic segmentation datasets, for example, ADE20k, is necessary. Since the idea of enhancing the segmentation performance with the help of both local and global attention (e.g. Swin Transformer) is not new, it is hard to explain the reason why the difference in attention patterns is more important in infrared semantic segmentation. The difference on attention pattern and their impact between RGB dataset and infrared dataset needs to be clarified.
2. As it is mentioned in line 475 “Excessive constraints on features may intensify the distillation shift and hinder the generalization of distilled models“, an ablation for distillation that combines the features and attention is needed. The comparison between its performance against each method individually can support the claim about excessive constraints.

**Questions:**

Since the proposed NMI-HAD is layer-wise, I’m curious about the performance when the distillation is more fine-grained. For example, why not apply the distillation on different heads? I may suggest the author to conduct an experiment comparing the performance between layer-wise approach and head-wise approach, which can have deeper insight on the choice of distillation strategy.

---

> ### Author Response · Authors · 2024-11-21
> **Response to Reviewer ctU1 (1/3)**
>
> Thanks for your effort in reviewing our paper and providing valuable feedback. We are pleased to hear your positive comments regarding our writing, experiments, and methods. We hope the following responses address your concerns effectively.
>
> > Q1: About the attention pattern in RGB and infrared datasets.
>
> Thanks for your appreciation of our analysis. This paper mainly discusses two insights about the attention maps in pre-trained ViT models: (1) The hybrid attention pattern is more important for semantic segmentation than other attention patterns; (2) The bias towards texture impairs the transfer performance on infrared semantic segmentation. The first insight is evident not only in infrared datasets but also in RGB and depth datasets, as mentioned in Lines 505-522 in the original paper. In contrast, the second insight is specific to the infrared modality, and this is where the attention maps have different impacts on RGB and infrared datasets. Further explanations are as follows:
>
> - **Insight1**: For experiments, Fig. 5 indicates the importance of the hybrid pattern in infrared datasets, while Tab. 10 of our paper demonstrates its effectiveness in RGB and depth segmentation datasets, including ADE20K. For visualization, we add the illustration of the attention maps on RGB image inputs in Fig. 13 of our revised draft. A comparison between Fig. 3 and Fig. 13 reveals that pre-trained models exhibit nearly identical distributions of attention patterns for both RGB and infrared images. **Therefore, we believe that the difference in attention patterns is important in both RGB and infrared semantic segmentation.** We will further highlight the scope of this insight in our final version. We further conduct the experiment about distilling on ImageNet using NMI-HAD and fine-tuning on ADE20K using LL-FPN, which also demonstrates the importance of hybrid attention patterns in RGB datasets. For more details on this experiment, please refer to our response to the Q1 of Reviewer FdMz.
> - **Insight2**: As stated in Sec. 3.4 of our paper, the bias towards texture would amplify the distribution shift and hinder the model’s generalization on infrared tasks, since texture information is scarce in infrared images. For example, MAE-Base outperforms DeiT-Base and DINO-Base on RGB datasets like ADE20K, while lags behind them on infrared datasets, as indicated in Tab. 2 of our paper. **Therefore, the texture bias of attention maps has different impacts on RGB and infrared datasets.**
>
> Additionally, compared to hierarchical architectures like the Swin Transformer which integrates both local and global attention mechanisms in their architectural design, this paper focuses on the pre-trained vanilla Vision Transformer (ViT), which does not incorporate such prior knowledge. Based on the thorough experiments and analysis, our study aims to convey that, despite the non-hierarchical nature of this architecture, different layers may exhibit distinct and hierarchical attention patterns, which have varying impacts on the model’s transfer performance. Furthermore, in light of these findings, we propose NMI-HAD and LL-FPN to leverage the significant influence of hybrid attention patterns on semantic segmentation tasks.

---

> ### Author Response · Authors · 2024-11-21
> **Response to Reviewer ctU1 (2/3)**
>
> > Q2: Ablation studies for distillation of combining the features and attention.
>
> Thanks for your valuable suggestion. We supplement the experiments in Tab. 6 of our paper by simultaneously using feature and attention distillation, as shown in the table below. Regardless of whether the distillation is performed using the 18th or 24th layer of the teacher model, the combined use of feature and attention distillation outperforms using features alone, but is inferior to using attention alone.
>
> From the perspective of feature constraints, feature distillation imposes restrictions on each token's features and inherently restricts the relationships between features, whereas attention distillation only constrains the relationships between features. In the context of this paper, the issue of distribution shift is relatively significant. Consequently, employing feature distillation makes the distilled model less capable of effectively adapting to new data distributions. In contrast, attention distillation only requires the preservation of the relationships between features, making it more suitable for knowledge distillation in scenarios with significant distribution shifts.
>
> Therefore, compared to using feature distillation alone, when both feature and attention distillation are employed together, the additional attention distillation objective reduces the emphasis on constraining each token's features during training, thereby overall decreasing the constraints on features. Conversely, compared to using attention distillation alone, the additional feature distillation objective increases the constraint on features, leading to a decline in performance.
>
> | Layer (Teacher) |    $\quad\quad\enspace$  Target    |   SODA    |  MFNet-T  | SCUT-Seg  |  Avg FT   |
> | :-------------: | :-------------: | :-------------: | :-------------: | :-------------: | :-------------: |
> | 18 (MAE-Large)  |    Feature only     |   65.66   |   48.44   |   66.55   |   60.22   |
> | 18 (MAE-Large)  |   Attention only    | **70.99** | **51.32** | **70.79** | **64.37** |
> | 18 (MAE-Large)  | Feature & Attention |   69.23   |   50.47   |   69.06   |   62.92   |
> | 24 (MAE-Large)  |    Feature only     |   66.86   |   49.00   |   66.61   |   60.82   |
> | 24 (MAE-Large)  |   Attention only    | **67.74** | **50.39** | **69.00** | **62.38** |
> | 24 (MAE-Large)  | Feature & Attention |   67.37   |   49.88   |   68.12   |   61.79   |

---

> ### Author Response · Authors · 2024-11-21
> **Response to Reviewer ctU1 (3/3)**
>
> > Q3: About the head-wise distillation.
>
> Thanks for your insightful suggestion. We think it is a valuable idea and conduct corresponding experiments as follows. The experimental setup involves using MAE-Large (16 attention heads for each layer) as the teacher to distill UNIP-Small (6 attention heads for each layer). First, we calculate the Normalized Mutual Information (NMI) for attention maps of each attention head in MAE-Large and observe that not all attention heads within the same layer exhibit the same attention pattern. Therefore, we categorize these attention heads into three patterns: local (l), hybrid (h), and global (g).
>
> We then select six attention heads (the total number of heads in UNIP-Small) as distillation targets and do not use the head misalignment strategy in Appendix B.4. For the 18th layer in MAE-Large, there are 5 global heads, 4 local heads, and 7 hybrid heads. We experiment with three different combinations: one containing only hybrid patterns (row 2), one containing only local and global patterns (row 3), and one containing all three patterns (row 4). The average NMI values for these combinations are comparable. Notably, **the combination containing only hybrid attention patterns achieves the best performance, demonstrating the effectiveness of hybrid attention patterns even in head-wise distillation.** Furthermore, using just 6 hybrid attention heads for distillation even surpasses the performance of distilling all 16 heads in the 18th layer (row 1). This phenomenon is also observed in the 24th layer. This suggests that there may be redundancy in the attention maps within a single layer. Therefore, we believe that more fine-grained distillation, such as head-wise distillation, is a highly promising research direction. We will discuss this in the future work section of our final version. We greatly appreciate your insightful suggestion and will continue to explore this avenue in the future.
>
> |      |   Method   | Layer (MAE-Large) |                   $\quad\quad\quad\quad\quad\quad$Target                    | Avg NMI |   SODA    |  MFNet-T  | SCUT-Seg  |  Avg FT   |
> | :--: | :--------: | :---: | :-----------------------------------------: | :-----: | :-------: | :-------: | :-------: | :-------: |
> |  1   | Layer-wise |  18   |                All 16 heads                 | 0.1185  | **70.99** |   51.32   |   70.79   |   64.37   |
> |  2   | Head-wise  |  18   | 5 (h), 6 (h), 8 (h), 10 (h), 13 (h), 15 (h) | 0.0985  |   70.37   | **52.01** | **71.82** | **64.73** |
> |  3   | Head-wise  |  18   |  2 (g), 3 (g), 4 (g), 7 (l), 9 (l), 14 (g)  | 0.1049  |   68.75   |   50.89   |   70.43   |   63.36   |
> |  4   | Head-wise  |  18   | 3 (g), 4 (g), 8 (h), 9 (l), 10 (h), 15 (h)  | 0.1077  |   70.07   |   51.65   |   69.76   |   63.83   |
> |  5   | Layer-wise |  24   |                All 16 heads                 | 0.1882  |   67.74   |   50.39   |   69.00   |   62.38   |
> |  6   | Head-wise  |  24   |  3 (h), 4 (h), 6 (h), 12(h), 15 (h), 16(h)  | 0.1092  | **69.95** | **51.82** | **69.88** | **63.88** |
>
> *The number(pattern) in the target column denotes the index of the selected head and the abbreviation of the corresponding pattern.*
>
> ---
>
> We would be more than happy to discuss any further questions!

---

> ### Comment · Area_Chair_PEMD · 2024-11-25
>
> Dear Reviewer ctU1,
>
> Could you kindly review the rebuttal thoroughly and let us know whether the authors have adequately addressed the issues raised or if you have any further questions.
>
> Best,
>
> AC of Submission2787

---

> ### Comment · Reviewer_ctU1 · 2024-11-25
>
> Thank you for your comment. Most of my concerns have been addressed, though, it appears to me that this paper does not have an emphasized motivation, which is also pointed out by reviewer FdMz. Although the key factor that affects the infrared semantic segmentation pretraining performance has been pointed out, I have to agree with the comment from FdMz that 'the proposed approach is not specific to infrared semantic segmentation tasks. After all, I believe this paper needs more analysis to emphasize the connection between the proposed NMI-HAD and infrared semantic segmentation. As a result, I decided to maintain my rating.

---

> > ### Author Response · Authors · 2024-11-26
> >
> > Thanks for your kindful feedback. We will further elaborate on the relationship between our motivations and methods in accordance with the paper.
> >
> > As stated in Lines 035-049 of the paper, our primary motivation is to enhance the performance of infrared semantic segmentation tasks. From the model perspective, factors affecting the performance on specific tasks include not only the design of the model architecture but also the quality of the model's pre-training. Previous works [1,2] have aimed to improve performance by designing specific network architectures for infrared semantic segmentation tasks. However, in the infrared domain, where labeled data is limited, the quality of the pre-trained model is also crucial. **Therefore, our work explores an alternative approach by emphasizing the optimization of pre-trained models specifically for infrared semantic segmentation tasks to enhance performance.** To facilitate this exploration, our work is organized into three stages: benchmark establishment, cause analysis, and method proposal, as illustrated in Fig. 1 of the paper.
> >
> > - **Benchmark Establishment (Sec. 2).** We establish a benchmark for the transfer performance of six RGB pre-training methods, encompassing a total of 18 pre-trained models, on three infrared semantic segmentation datasets (Sec. 2.1). Our findings reveal several key phenomena (Sec. 2.2), such as the lack of correlation between model performance on ImageNet and infrared segmentation datasets (Tab. 1), and the superior generalization of supervised and contrastive learning methods over masked image modeling methods in the context of infrared segmentation tasks (Fig. 2).
> >
> > - **Cause Analysis (Sec. 3).** To analyze the performance discrepancies among various pre-training methods in infrared segmentation tasks, we conduct an in-depth analysis of the attention maps from the pre-trained models. Our findings indicate that the degree of focus on local and global information (Sec. 3.1 - Sec. 3.3), as well as on shape and texture information (Sec. 3.4), significantly impacts the performance of infrared semantic segmentation tasks. We further validate through corresponding experiments that the existence of hybrid attention patterns (Fig. 3 - Fig. 5) and the reduced bias towards texture (Tab. 2) both play crucial roles in enhancing the performance of pre-trained models in infrared segmentation tasks.
> >
> > - **Method Proposal (Sec. 4).** Based on the observations and analyses from the previous two sections, we propose UNIP, a framework designed to improve the infrared segmentation performance of pre-trained models through three key aspects: the pre-training objective (NMI-HAD), the pre-training data (InfMix), and the fine-tuning architecture (LL-FPN). Both NMI-HAD and LL-FPN enhance performance by effectively leveraging hybrid attention patterns, while InfMix enhances performance by reducing the pre-trained model's bias toward texture information. **Importantly, our approach does not alter the structure of the backbone model or the decoder; instead, we focus on targeted pre-training specifically designed for infrared segmentation tasks. We believe this is one way in which the proposed method is specific to infrared semantic segmentation tasks.** As a result, our pre-trained models significantly outperform RGB pre-trained models of comparable or even larger sizes in infrared semantic segmentation tasks (Tab. 4), and also achieve superior performance compared to other models specifically designed for infrared segmentation (Tab. 5).
> >
> > Thank you again for your valuable suggestions. We are open to your further response and instructions.
> >
> > ---
> >
> > [1] *Chen J, Bai X. Atmospheric transmission and thermal inertia induced blind road segmentation with a large-scale dataset tbrsd. ICCV, 2023.*
> >
> > [2] *Li C, Xia W, Yan Y, et al. Segmenting objects in day and night: Edge-conditioned CNN for thermal image semantic segmentation. TNNLS, 2021.*

---

> > > ### Comment · Reviewer_ctU1 · 2024-11-27
> > >
> > > Thank you for your comment. The analysis of the relationship between the motivation and the method is clear in this comment. I strongly suggest the author add them to the paper in revision or in the camera-ready version, which will make this paper more persuasive.

---

> > > > ### Author Response · Authors · 2024-11-28
> > > >
> > > > Dear Reviewer ctU1,
> > > >
> > > > Thanks for your valuable suggestion. We have uploaded a newly revised draft with modifications marked in purple and added the analysis of the relationship between the motivation and the method in Appendix B (Lines 805-839). We have also revised several expressions in the introduction (Sec. 1, Lines 044 and 049) to make our motivations more explicit. Additionally, we supplement the experiments about the head-wise distillation in Appendix E (Lines 1176-1187, 1223-1226).
> > > >
> > > > We sincerely hope that we have addressed your concerns, and we would greatly appreciate knowing if you might be willing to adjust your evaluation of our paper in light of our responses.
> > > >
> > > > Best regards,
> > > >
> > > > The Authors

---

### Author Response · Authors · 2024-11-21
**Summary of Revisions**

We sincerely thank all reviewers for your valuable comments. We upload a newly revised draft with several modifications **marked in purple**, detailed as follows:

- Change the title from “… Semantic Segmentation” to “… Infrared Semantic Segmentation”.
- Discuss the relationship between our motivations and methods. (Appendix B, Lines 805-839)
- Add the experiments about the two-stage pre-training. (Appendix E, Lines 1159-1169)
- Add the experiments about head-wise distillation. (Appendix E, Lines 1176-1187, Lines 1223-1226)
- Add a visualization of the attention maps for RGB image inputs in Fig. 13. (Appendix G, Lines 1359-1396)
- Replace “SSL” with “pre-training” and include appropriate citations. (Sec. 2.1, Line 146)
- Align the letters of ViT in Fig.8. (Appendix C.2, Lines 918-933)

We hope our response addresses your concerns. We would be more than happy to discuss any further questions!

---

### Meta-Review · Area_Chair_PEMD · 2024-12-18

**Metareview:**

(a) This paper explores the role of pre-trained attention patterns in infrared semantic segmentation, proposing the new pre-training framework termed NMI-HAD and LL-FPN to enhance hybrid attention patterns and improve fine-tuning performance.

(b) Strengths: This work is well-written, easy to follow, and convincingly motivated. It emphasizes the importance of pre-training methods in semantic segmentation and introduces NMI to identify attention patterns for distillation, contributing to the community. Extensive experiments demonstrate that the proposed approach significantly outperforms state-of-the-art infrared and RGB segmentation methods.

(c) Weaknesses: In the initial version, the title is inappropriate by ignoring the keyword ``infrared''. It may lack comparisons against domain-adaptive methods, more pretraining strategies, the clarification about which layer to distill, and unclear analysis of the relationship between the motivation and the method. The authors have demonstrated sufficient explanations during the rebuttal.

(d) The most important reasons for acceptance are the authors have conducted an in-depth analysis of the attention maps for pretraining and identified the hybrid attention patterns beneficial for the task. These findings offer good insights for the community. Furthermore, the relationship between the motivation and methods is clearly articulated, with step-by-step research insights that are highly promising. Although two reviewers (FdMz and nH3J) initially gave scores of 5, the AC carefully reviewed the rebuttal discussions and found that the authors successfully addressed all their concerns. Since these reviewers did not provide a final response, we consider the feedback to be entirely positive and recommend the paper for acceptance.

**Additional Comments On Reviewer Discussion:**

(a) Reviewer ctU1 shows concerns about the need for more comparison of attention patterns with RGB datasets like ADE20k, clarification on why differences in attention patterns are particularly significant for infrared segmentation, and an ablation study combining features and attention to validate the claim about excessive constraints in distillation. The authors have addressed all the concerns during several rounds of discussions.

(b) Reviewer FdMz finds that the motivation of the paper is unclear, as it alternates between emphasizing infrared segmentation performance and the method's versatility. They note that the approach is not specific to infrared tasks and lacks experiments on general datasets and comparisons with related works in visual pre-training and transfer learning. While the method's contribution to selecting layers for distillation is acknowledged, concerns are raised about the terminology and effectiveness of using "local-global" and "global" attention patterns, as the latter represents attention collapse. Additionally, they question the conclusion about the last layer being the best distillation target, suggesting it may differ when using hierarchical features. Most of the concerns are addressed, however, the reviewer still feels that the methods proposed in this work or the conclusions drawn from the analysis are not unique to infrared images. Since the reviewer fails to give the final response, the AC has carefully checked more context about this issue and discovered that the authors have made it very clear by depicting the three stages of research pipeline: 1) benchmark establishment; 2) causal analysis; 3) method proposal.

(c) Reviewer nH3J shows that the paper's title is misleading, as it suggests a general semantic segmentation task while focusing on infrared segmentation of small models. They question the effectiveness of using an MAE-based teacher model for distillation and note the lack of comparisons with existing distillation strategies in domain-adaptive methods. Additionally, they recommend experiments on combined datasets, such as training on RGB images and fine-tuning on infrared, or joint training, to further validate the method. The authors have resolved most issues, but the reviewer still believes there are some conflicts between the claim that "the MIM (Masked Image Modeling) strategy focuses more on texture information and is not suitable for segmenting infrared images" and the method using MAE-L as the teacher. Since the reviewer fails to give the final response to this, the AC has carefully examined the details about the issue. It turns out that the performance evaluation supports the authors.

(d) Reviewer gRqm finds that there are no MAJOR weaknesses and suggests for minor revisions. The authors have updated these modifications.

---

### Decision · Program_Chairs · 2025-01-22

Accept (Poster)